# INFORMATION THEORETIC TEXT-TO-IMAGE ALIGNMENT

**Chao Wang**[1,2]**, Giulio Franzese**[1]**, Alessandro Finamore**[2]**, Massimo Gallo**[2]**, Pietro Michiardi**[1]
EURECOM[1], Huawei Technologies SASU, France[2]
[1]`{chao.wang, giulio.franzeze, pietro.michiardi}@eurecom.fr`
[2]`{wang.chao3, alessandro.finamore, massimo.gallo}@huawei.com`

## ABSTRACT

Diffusion models for Text-to-Image (T2I) conditional generation have recently achieved tremendous success. Yet, aligning these models with user's intentions still involves a laborious trial-and-error process, and this challenging alignment problem has attracted considerable attention from the research community. In this work, instead of relying on fine-grained linguistic analyses of prompts, human annotation, or auxiliary vision-language models, we use Mutual Information (MI) to guide model alignment. In brief, our method uses self-supervised fine-tuning and relies on a point-wise MI estimation between prompts and images to create a synthetic fine-tuning set for improving model alignment. Our analysis indicates that our method is superior to the state-of-the-art, yet it only requires the pre-trained denoising network of the T2I model itself to estimate MI, and a simple fine-tuning strategy that improves alignment while maintaining image quality. Code available at https://github.com/Chao0511/mitune.

## 1 INTRODUCTION

Generative models used for Text-to-Image (T2I) conditional generation (Rombach et al., 2022; Ramesh et al., 2022; Saharia et al., 2022; Balaji et al., 2022b; Gafni et al., 2022; Podell et al., 2024) have reached impressive performance. In particular, diffusion models (Song & Ermon, 2019b; Ho et al., 2020; Kingma et al., 2021; Song & Ermon, 2020; Song et al., 2021; Dhariwal & Nichol, 2021) generate extremely high-quality images by specifying a natural text prompt that acts as a guiding signal (Ho & Salimans, 2022; Nichol et al., 2022; Rombach et al., 2022). Yet, accurately translating prompts into images with the intended semantics is still complex (Conwell & Ullman, 2022; Feng et al., 2023a; Wang et al., 2023a). Issues include catastrophic neglecting (i.e., prompt elements are not generated), incorrect attribute binding (i.e., elements attributes such as color, shape, and texture are missing or wrongly assigned), incorrect spatial layout (i.e., elements are not correctly positioned), and a general difficulty in handling complex prompts (Wu et al., 2024).

On the one hand, quantifying *model alignment* is not trivial. Various works (Hu et al., 2023; Gordon et al., 2023; Grimal et al., 2024) propose different metrics, most of which use complementary Visual Question Answering (VQA) models or Large Language Models (LLMs) to create scores measuring and explaining alignment. Moreover, a recent work (Huang et al., 2023) introduces a comprehensive benchmark suite to ease comparison among different metrics and modeling techniques via "categories", i.e., a pre-defined set of attribute binding, spatial-related, and other tasks.

On the other hand, addressing T2I model alignment is even more challenging than measuring it. Broadly, we can group the related literature into two main families: *inference-time* and *fine-tuning* methods. For inference-time methods, the key intuition is that the generative process can be optimized by modifying the reverse path of the latent variables. Some works (Chefer et al., 2023a; Li et al., 2023b; Rassin et al., 2023) mitigate failures by refining the cross-attention units (Tang et al., 2023) of the denoising network of

Stable Diffusion (SD) (Rombach et al., 2022) on-the-fly, ensuring they attend to all subject tokens in the prompt (typically directly specified as a complementary prompt-specific input for the alignment process) and strengthen their activations. Other inference-time methods (Agarwal et al. (2023); Liu et al. (2022); Kang et al. (2023); Dahary et al. (2024); Meral et al. (2024); Feng et al. (2023b); Kim et al. (2023); Wu et al. (2023a); Zhang et al. (2024a;b)), focus on individual failure cases. These approaches ($i$) require a linguistic analysis of prompts, leading to specialized solutions that rely on auxiliary models for prompt understanding, and ($ii$) result in considerably longer image generation time due to extra optimization costs during sampling.

Considering fine-tuning methods, some works (Wu et al., 2023d; Lee et al., 2023) require human annotations to prepare a fine-tuning set, while others (Fan et al., 2023; Wallace et al., 2023; Clark et al., 2024) rely on Reinforcement Learning (RL), Direct Preference Optimization (DPO), or a differentiable reward function to steer model behavior. Recent methods use self-playing (Yuan et al., 2024; Xu et al., 2023; Sun et al., 2023; Wang et al., 2023b; Ma et al., 2023), auxiliary models such as VQA (Li et al., 2023a; Jiang et al., 2024) or segmentation maps (Kirillov et al., 2023) in a semi-supervised fine-tuning setting. While these methods do not introduce extra inference time costs, they still require human annotation (which is subjective, costly, and does not scale well) and/or auxiliary models to guide the fine-tuning.

Complementary to both families are *heuristic*-based methods that rely on a variety of "tricks", such as prompt engineering (Witteveen & Andrews, 2022; Liu & Chilton, 2022; Wang et al., 2023a), negative prompting (hfn, 2022; Mahajan et al., 2023; Ogezi & Shi, 2024), prompt rewriting (Mañas et al., 2024) or brute force an appropriate seed selection (Samuel et al., 2024; Karthik et al., 2023). While these methods can be beneficial in specific cases, they fundamentally shift the alignment problem to users.

Overall, current approaches require extra information (human input, auxiliary models, and additional data). To the best of our knowledge, no previous work investigates *self-supervised* approaches for T2I alignment, i.e., the use of a pre-trained model to generate images given a specific set of prompts, and select the most aligned ones to prepare a fine-tuning set, without using auxiliary models. In this work, we investigate this strategy from an information theoretic perspective, by using MI to quantify the non-linear prompt-image relationship. In particular, we focus on the estimation of *point-wise* MI using neural estimators (Belghazi et al., 2018; Song & Ermon, 2019a; Brekelmans et al., 2022; Franzese et al., 2024; Kong et al., 2024), and study if and how MI can be used as a meaningful signal to improve T2I alignment, without relying on linguistic analysis of prompts, nor auxiliary models or heuristics. Our method unfolds as follows. We build upon the work in (Franzese et al., 2024) and extend it to compute point-wise MI. We then proceed with a self-supervised fine-tuning approach, whereby we use point-wise MI to construct a fine-tuning set using synthetic data generated by the T2I model itself. We then use the recent adapter presented in (Liu et al., 2024) to fine-tune a small fraction of weights injected in the T2I model denoising network. In summary, our work presents the following contributions:

**(1) We define a point-wise MI estimator suitable for a discrete-time setting** (§ 2). We empirically study whether MI between natural prompts and corresponding images considering both qualitative and quantitative approaches. Specifically, we show that MI provides a meaningful indication of alignment with respect to both alignment metrics (BLIP-VQA and HPS) as well as a users study (§ 3.1).

**(2) We design a self-supervised fine-tuning approach**, called MI-TUNE (§ 3.2), that uses a small number of fine-tuning samples to align a pre-trained T2I model without extra auxiliary models or inference overhead.

**(3) We perform an extensive experimental campaign** using a recent T2I benchmark suite (Huang et al., 2023) and SD-2.1-base as base model obtaining sizable improvement compared to six alternative methods (§ 4). Those benefits hold also when considering more complex tasks (based on DiffusionDB (Wang et al., 2022)) and alternative base models (namely, SDXL (Podell et al., 2024)). Moreover we study the trade-off between T2I alignment and image quality that has been overlooked in the literature. Specifically, while the well-known FID, FD-DINO and CMMD metrics suggest a modest image quality/variety deterioration as a

consequence of alignment objectives, optimizing the Classifier Free Guidance (CFG) hyper-parameter of the fine-tuned model at generation time, enables finding a "sweet spot" between T2I alignment and image quality.

## 2 PRELIMINARIES

**Diffusion models.** Denoising diffusion models (Ho et al., 2020; Sohl-Dickstein et al., 2015) are generative models characterized by a forward process, that is fixed to a Markov chain that gradually adds Gaussian noise to the data according to a carefully selected variance schedule $\beta_t$, and a corresponding discrete-time reverse process, that has a Markov structure as well. Intuitively, diffusion models rely on the principle of iterative denoising: starting from a simple distribution $\boldsymbol{x}_T \sim \mathcal{N}(\boldsymbol{0}, \boldsymbol{I})$, samples are generated by iterative applications of a denoising network $\boldsymbol{\epsilon}_{\boldsymbol{\theta}}$, that removes noise over $T$ denoising steps. A simple way to learn the denoising network $\boldsymbol{\epsilon}_{\boldsymbol{\theta}}$ is to consider a re-weighted variational lower bound of the marginal likelihood:

$$\mathcal{L}_{\text{simple}}(\boldsymbol{\theta}) = \mathbb{E}_{t \sim U(0,T), \boldsymbol{x}_0 \sim p_{\text{data}}, \boldsymbol{\epsilon} \sim \mathcal{N}(\boldsymbol{0}, \boldsymbol{I})} \left[ ||\boldsymbol{\epsilon} - \boldsymbol{\epsilon}_{\boldsymbol{\theta}}(\sqrt{\bar{\alpha}_t}\boldsymbol{x}_0 + (\sqrt{1 - \bar{\alpha}_t})\boldsymbol{\epsilon}, t)||^2 \right], \quad (1)$$

where $\alpha_t = 1 - \beta_t$, $\bar{\alpha}_t = \prod_{s=1}^{t} \alpha_s$. For sampling, we let $\sigma_t^2 = \beta_t$. A similar variational objective can be obtained by switching perspective from discrete to continuous time (Song et al., 2021), whereby the denoising network approximates a score function of the data distribution. For image data, the denoising network is typically parameterized by a UNET (Ronneberger et al., 2015; Rombach et al., 2022).

This simple formulation has been extended to conditional generation (Ho & Salimans, 2021), whereby a conditioning signal $\boldsymbol{p}$ injects "external information" in the iterative denoising process. This requires a simple extension to the denoising network such that it can accept the conditioning signal: $\boldsymbol{\epsilon}_{\boldsymbol{\theta}}(\boldsymbol{x}_t, \boldsymbol{p}, t)$. Then, during training, a randomized approach allows to learn both the conditional and unconditional variants of the denoising network, for example by assigning a null value to the conditioning signal. At sampling time, a weighted linear combination of the conditional and unconditional networks, such as $\tilde{\boldsymbol{\epsilon}}_{\boldsymbol{\theta}}(\boldsymbol{x}_t, \boldsymbol{p}, t) = \boldsymbol{\epsilon}_{\boldsymbol{\theta}}(\boldsymbol{x}_t, \emptyset, t) + \gamma(\boldsymbol{\epsilon}_{\boldsymbol{\theta}}(\boldsymbol{x}_t, \boldsymbol{p}, t) - \boldsymbol{\epsilon}_{\boldsymbol{\theta}}(\boldsymbol{x}_t, \emptyset, t))$ can be used.

In this work, we use pre-trained latent diffusion models operating on a learned projection of the input data $\boldsymbol{x}_0$ into a corresponding latent variable $\boldsymbol{z}_0$ which is lower-dimensional compared to the original data. Moreover, the conditioning signal $\boldsymbol{p}$ is obtained by a text encoder such as CLIP (Radford et al., 2021).

**MI estimation.** MI is a central measure to study the non-linear dependence between random variables (Shannon, 1948; MacKay, 2003), and has been extensively used in machine learning for representation learning (Bell & Sejnowski, 1995; Stratos, 2019; Belghazi et al., 2018; Oord et al., 2018; Hjelm et al., 2019), and for both training (Alemi et al., 2016; Chen et al., 2016; Zhao et al., 2018) and evaluating generative models (Alemi & Fischer, 2018; Huang et al., 2020).

For many problems of interest, precise computation of MI is not trivial (McAllester & Stratos, 2020; Paninski, 2003). Consequently, a wide range of techniques for MI estimation have flourished. In this work, we focus on realistic and high-dimensional data, which calls for recent advances in MI estimation (Papamakarios et al., 2017; Belghazi et al., 2018; Oord et al., 2018; Song & Ermon, 2019a; Rhodes et al., 2020; Letizia & Tonello, 2022; Brekelmans et al., 2022; Kong et al., 2024). In particular, we capitalize on a recent method (Franzese et al., 2024), that relies on the theory behind continuous-time diffusion processes (Song et al., 2021) and uses the Girsanov Theorem (Øksendal, 2003) to show that score functions can be used to compute the Kullback-Leibler (KL) divergence between two distributions. In what follows, we use a simplified notation and gloss over several mathematical details to favor intuition over rigor. Here we consider discrete-time diffusion models, which are equivalent to the continuous-time counterpart under the variational formulation, up to constants and discretization errors (Song et al., 2021).

We begin by considering the two arbitrary random variables $\boldsymbol{z}$ and $\boldsymbol{p}$ which are sampled from the joint distribution $p_{\text{latent,prompt}}$, where the former corresponds to the distribution of the projections in a latent space

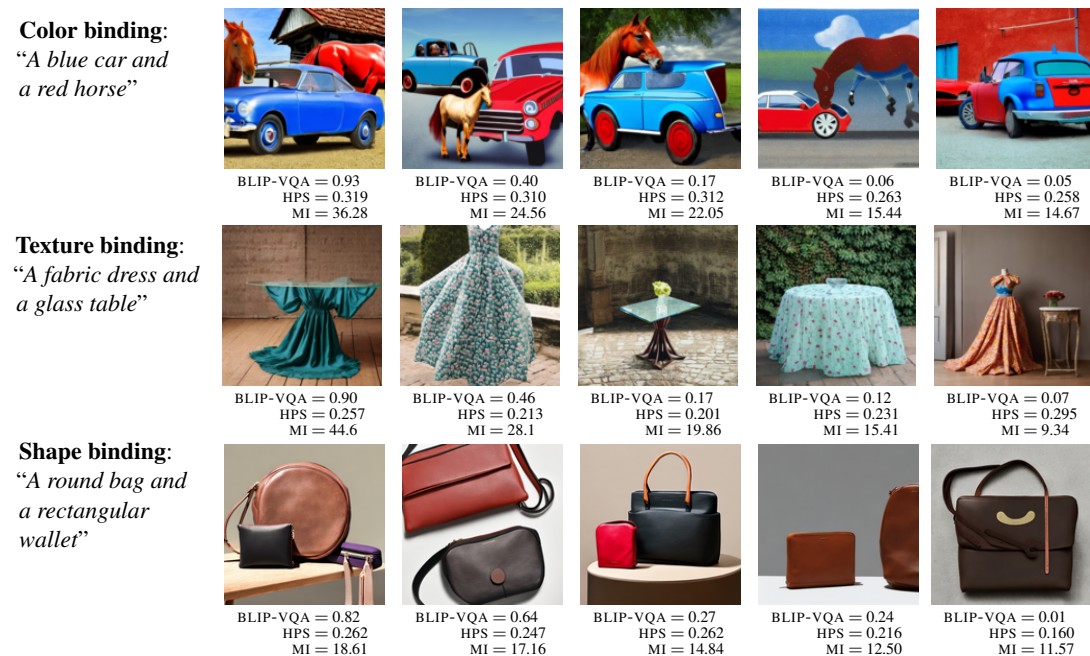

**Figure 1:** Qualitative analysis of MI as an alignment measure (all metrics decrease from left to right). See also Appendix I.

of the image distribution, and the latter to the distribution of prompts used for conditional generation. Then, following the approach in (Franzese et al., 2024), with the necessary adaptation to the discrete domain (see Appendix A for details), point-wise MI estimation can be obtained as follows:

$$\mathbf{I}(\boldsymbol{z}, \boldsymbol{p}) = \mathbb{E}_{t, \boldsymbol{\epsilon} \sim \mathcal{N}(\boldsymbol{0}, \boldsymbol{I})} \left[ \kappa_t || \boldsymbol{\epsilon_\theta}(\boldsymbol{z}_t, \boldsymbol{p}, t) - \boldsymbol{\epsilon_\theta}(\boldsymbol{z}_t, \emptyset, t) ||^2 \right], \quad \kappa_t = \frac{\beta_t T}{2\alpha_t (1 - \bar{\alpha}_t)}. \tag{2}$$

Given a pre-trained diffusion model, we compute an expectation (over diffusion times $t$) of the scaled squared norm of the difference between the conditional $\boldsymbol{\epsilon_\theta}(\boldsymbol{z}_t, \boldsymbol{p}, t)$ and unconditional networks $\boldsymbol{\epsilon_\theta}(\boldsymbol{z}_t, \emptyset, t)$, which corresponds to an estimate of the point-wise MI between an image and a prompt. Intuitively, the difference between these scores quantifies how much extra knowledge of the prompt helps in denoising the perturbed images. This is both a key ingredient and a competitive advantage of our method, as it enables a self-contained approach to alignment based on the T2I model alone without auxiliary models or human feedback.

## 3 OUR METHOD: MI-TUNE

The T2I alignment problem arises when user's intentions, as expressed through natural text prompts, fail to materialize in the generated image. Our novel approach aims to address alignment using a theoretically grounded MI estimation, that applies across various contexts. To improve model alignment, we introduce a self-supervised fine-tuning method. Leveraging the T2I model itself, we estimate MI and generate an information-theoretic enhanced fine-tuning dataset. While our focus in this work is on T2I alignment, our framework remains extensible to other modalities.

### 3.1 IS MUTUAL INFORMATION MEANINGFUL FOR ALIGNMENT?

To the best of our knowledge, MI has never been evaluated as a *meaningful signal* for T2I alignment. As such, in this section we perform both qualitative and quantitative analyses to investigate this aspect.

**Qualitative analysis.** Starting with a qualitative analysis, we select a set of simple prompts to probe color, texture, and shape attribute binding from T2I-CompBench (Huang et al., 2023) using SD (Rombach et al., 2022) (specifically SD-2.1-base) to generate the corresponding images. We then measure the well-known BLIP-VQA (Huang et al., 2023) and Human Preference Score (HPS) (Wu et al., 2023b) alignment metrics as well as point-wise MI estimates. BLIP-VQA uses a large vision-language model to compute an alignment score, by casting questions against an image to verify that the prompt used to generate it is well represented. HPS is an elaborate metric that uses an auxiliary pre-trained model, blending alignment with aesthetics according to human perception, which are factors that can sometimes be in conflict. Figure 1 collects some examples and related metric scores revealing a substantial agreement among all measures: all metrics decrease from left to right in the figure, as prompt-image alignment deteriorates.

**Quantitative analysis.** To quantitatively measure the agreement between MI and well-established alignment metrics, we use all 700 prompts from T2I-CompBench and use SD (again, SD-2.1-base) to generate 50 images per prompt. We use point-wise MI to rank such images and select the 1st, 25th, and 50th. For these three representative images, we compute BLIP-VQA and HPS scores and re-rank them according to both metrics. Last, we measure agreement between the three rankings using Kendall's $\tau$ method (Kendall, 1938), and average results across all prompts. Results indicate good agreement between MI and BLIP-VQA ($\tau = 0.4$), and a strong agreement between MI and HPS ($\tau = 0.68$).

To strengthen our analysis, we also perform a users study eliciting human preference (see Appendix B.1 for details). Given a randomly selected prompt from T2I-CompBench that users can read, we present the top-ranked generated image (among the 50) according to MI, BLIP-VQA and HPS, in a randomized order. Users can select one or more images to indicate their preference regarding alignment and aesthetics, for a total of 10 random prompts per user. From the 102 surveys from 46 users, we find that human preference for prompt-image pairs goes to MI for 69.1%, BLIP-VQA for 73.5% and HPS for 52.2% of the cases, respectively.

**Relevant literature.** Overall, our analyses support our intuition by which MI *is a meaningful signal for alignment* (and possibly aesthetics too), setting the stage for our T2I alignment method. Our intuition is also supported by recent studies investigating the information flow in the generative process of diffusion models. Specifically, Kong et al. (2024) estimates pixel-wise mutual information between natural prompts and the images generated at each time-step of a backward diffusion process. They compare such "information maps" to cross-attention maps (Tang et al., 2023) in an experiment involving prompt manipulation – modifications of the initial prompt during reverse diffusion – and conclude that MI is much more sensitive to information flow from prompt to images. In a similar vein, Franzese et al. (2024) compute MI between prompt and images at different stages of the reverse process of image generation. Experimental evidence indicates that MI can be used to analyze various reverse diffusion phases: noise, semantic, and denoising stages (Balaji et al., 2022a). While previous studies do not explicitly focus on alignment, they *indirectly* support our intuition that MI estimated using a diffusion model gauges the amount of information a text prompt conveys about an image (and vice-versa) which is key for T2I alignment.

### 3.2 SELF-SUPERVISED FINE-TUNING WITH MI-TUNE

In summary, given a pre-trained diffusion model such as SD (Rombach et al., 2022) or any variant, such as SDXL (Podell et al., 2024), we leverage our point-wise MI estimation method to select a small fine-tuning dataset set of information-theoretic aligned examples.

Our self-supervised alignment method **relies on the pre-trained model only** to produce a given amount of fine-tuning data, which is then filtered to retain prompt-image pairs with a high degree of alignment, according

to pair-wise MI **estimates obtained using only the pre-trained model**. We begin with a set of fine-tuning prompts $\mathcal{P}$, which can be either manually crafted, or borrowed from available prompt collections (Wang et al., 2023a; Huang et al., 2023). Ideally, fine-tuning prompts should be conceived to stress the pre-trained model with challenging attribute and spatial bindings, or complex rendering tasks.

---

**Algorithm 1:** MI-TUNE

**Input** : Pre-trained model: $\boldsymbol{\epsilon}_\theta$, Prompt set: $\mathcal{P}$
**Hyper par** : Image pool size: $M$; Top MI-aligned samples: $k$
**Output** : Fine-tuned diffusion model $\boldsymbol{\epsilon}_{\theta*}$

   // Fine-tuning set
1  $\mathcal{S} \leftarrow [\,]$
2  **for** $\boldsymbol{p}^{(i)}$ *in* $\mathcal{P}$ **do**
3     **for** $j \in \{1, \cdots, M\}$ **do**
      // Generate and compute MI
4       $\boldsymbol{z}^{(j)}, \mathrm{I}(\boldsymbol{z}^{(j)}, \boldsymbol{p}^{(i)}) = \texttt{PointWiseMI}(\boldsymbol{\epsilon}_\theta, \boldsymbol{p}^{(i)})$
      // Append samples and MI
5       $\mathcal{S}[\boldsymbol{p}^{(i)}].\texttt{append}(\boldsymbol{z}^{(j)}, \mathrm{I}(\boldsymbol{z}^{(j)}, \boldsymbol{p}^{(i)}))$
6     **end**
    // Retain only Top-$k$ elements
7     $\mathcal{S}[\boldsymbol{p}^{(i)}] = \texttt{Top-}k(\mathcal{S}[\boldsymbol{p}^{(i)}])$
8  **end**
9  **return** $\boldsymbol{\epsilon}_{\theta*} = \texttt{FineTune}(\boldsymbol{\epsilon}_\theta, \mathcal{S})$

---

**Algorithm 2:** Point-wise MI Estimation

**Input** : Pre-trained model: $\boldsymbol{\epsilon}_\theta$; Prompt: $\boldsymbol{p}$
**Output** : Generated latent: $\boldsymbol{z}$; Point-wise MI: $\mathrm{I}(\boldsymbol{z}, \boldsymbol{p})$

1  **Function** $\texttt{PointWiseMI}(\boldsymbol{\epsilon}_\theta, \boldsymbol{p})$:
    // Initial latent sample
2     $\boldsymbol{z}_T \sim \mathcal{N}(\mathbf{0}, \boldsymbol{I})$ **for** $t$ *in* $T, ..., 0$ **do**
      // MI estimation (Eq. (2))
3       $\mathrm{I}(\boldsymbol{z}_t, \boldsymbol{p}) \mathrel{+}=$
      $\left[\kappa_t || \boldsymbol{\epsilon}_\theta(\boldsymbol{z}_t, \boldsymbol{p}, t) - \boldsymbol{\epsilon}_\theta(\boldsymbol{z}_t, \emptyset, t) ||^2\right]$
      // Noise sample
4       $\boldsymbol{w} \sim \mathcal{N}(\mathbf{0}, \boldsymbol{I})$ if $t > 1$, else $\boldsymbol{w} = \mathbf{0}$
      // Sampling step
5       $\boldsymbol{z}_{t-1} =$
      $\frac{1}{\sqrt{\alpha_t}}\left(\boldsymbol{z}_t - \frac{1-\alpha_t}{\sqrt{1-\bar{\alpha}_t}}\boldsymbol{\epsilon}_\theta(\boldsymbol{z}_t, \boldsymbol{p}, t)\right) + \sigma_t \boldsymbol{w}$
6     **end**
7     **return** $\boldsymbol{z}, \mathrm{I}(\boldsymbol{z}, \boldsymbol{p})$

---

As described in Algorithm 1, for each prompt $p^{(i)}$ in the fine-tuning set $\mathcal{P}$, we use the pre-trained model to generate a fixed number $M$ of synthetic images. Given prompt-image pairs $(p^{(i)}; z^{(j)})$, $j \in [1, M]$, we estimate pair-wise MI and select the top $k$ pairs, which will be part of the model fine-tuning dataset $\mathcal{S}$. Finally, we augment the pre-trained model with adapters (Hu et al., 2021; Liu et al., 2024), and proceed with fine-tuning. We study the impact of the adapter choice, and whether only the denoising network or both the denoising and text encoder networks should be fine-tuned (Appendix E.2). Moreover, we measure the impact of the number of fine-tuning rounds $R$ to the pre-trained model, i.e., we renew the fine-tuning dataset $\mathcal{S}$ using the fine-tuned model, and re-fine-tune it using Algorithm 1 (§ 4.4). Our efficient implementation combines latent generation and point-wise MI computation as shown in Algorithm 2. Since MI estimation involves computing an expectation over diffusion times $t$, it is easy to combine generation and estimation in the same loop. Moreover, the function is easy to parallelize to significantly speed up the fine-tuning set $\mathcal{S}$ composition.

# 4 EXPERIMENTAL EVALUATION

## 4.1 BENCHMARK AND METRICS

**Benchmark.** We compare all techniques using T2I-CompBench (Huang et al., 2023), a benchmark composed of 700/300 (train/test) prompts across 6 *categories* including attribute binding (color, shape, and texture categories), object relationships (2D-spatial and non-spatial associations), and complex composition tasks. These prompts were generated with predefined rules or ChatGPT (OpenAI, 2024). We also assess MI-TUNE performance on more realistic prompts by sampling 5,000/1,250 (train/test) prompt-image pairs from DiffusionDB (Wang et al., 2022), a large-scale dataset composed of complex human-crafted prompts paired with the corresponding images generated from a SD model.

**Alignment Metrics.** Evaluating T2I alignment is difficult as it requires a detailed understanding of prompt-images pairs, and many metrics have been proposed, e.g., CLIP (Hessel et al., 2021; Radford et al., 2021), MINIGPT-4 (Zhu et al., 2024), and human evaluation. In our work we use BLIP-VQA (Huang et al., 2023), HPS (Wu et al., 2023c) and UniDet (Zhou et al., 2022). While BLIP-VQA computes a score with a questions-answers approach – a given prompt is decomposed and each part is transformed into a question for an auxiliary VQA model; then, answers are aggregated into a single score – only based on alignment, HPS includes both alignment and aesthetics – this is enabled by an auxiliary model pre-trained using human-annotated data. As in (Huang et al., 2023), the 2D-spatial category is evaluated using the UniDet object detection model.

We complement these metrics with a user study. We randomly select 100 prompts per category, and generate 10 pictures per prompt for each method we consider in our evaluation. Then, we run surveys composed of 12 rounds (2 for each category), each showing to the user a randomly selected prompt and a randomly selected image for each method, randomly arranged in a grid. At each round, users need to select zero or more images they consider aligned with the prompt. Overall, we collected 42 surveys from 5 users, from which we computed the total percentage of times each method was selected for each category (Appendix B.2).

**Image quality metrics.** Assessing performance only considering alignment metrics can hide undesired effects. Intuitively, a strong adherence to a given prompt reduces the generative process "degrees of freedom" and this trade-off might not be visible even to a trained eye. To investigate these dynamics we compute FID (Heusel et al., 2017), FD-DINO (Oquab et al., 2024) and CMMD (Jayasumana et al., 2024) scores – FID favors natural colors and textures but struggles to detect objects/shapes distortion, while FD-DINO and CMMD favor image content. Following (Imagen-Team et al., 2024), rather than using the T2I-CompBench test set, we compute the metrics using 30k samples of the MS-COCO-2014 (Lin et al., 2015) validation set.

## 4.2 MI-TUNE FINE-TUNING

**Base models.** We mainly run our benchmark using SD-2.1-base as base model, but we also report results of the application of MI-TUNE on SDXL to demonstrate its flexibility.

**Fine-tuning sets.** T2I-CompBench contains 700 training prompts for each category. When using MI-TUNE, we generate $M = 50$ images for each prompt using the pre-trained model, compute their point-wise MI, and select the top $k = 1$[1] (sensitivity to $M$ and $k$ in Appendix E.1). For the 2D-Spatial category, we also compose fine-tuning sets generating images from SPRIGHT (Chatterjee et al., 2024) – a model optimized for this (more challenging) category and fine-tuned from SD-2.1 (a higher resolution version of SD-2.1-base). Last, we also contrast MI-TUNE fine-tuning set composition against ($i$) using HPS rather than MI for image selection,[2] ($ii$) using both MI-selected and real-pictures and ($iii$) images from DiffusionDB.

**Fine-tuning weights.** In our work, fine-tuning corresponds to injecting DoRA (Liu et al., 2024) adapters (rank and scaling factor $\alpha$ are set to 32) only into the attention layers and fully connected layers of the denoising UNET network, whereas other layers are frozen.[3]

**Other hyperparams search.** We consider up to $R \in [1, 3]$ rounds of fine-tuning i.e., using as base model the one obtained from previous round and apply Algorithm 1, and Classifier Free Guidance (CFG) $\in [2.5, 7.5]$. For each fine-tuned model we then compute all alignment and image quality metrics. More fine-grained hyperparams details and computational costs considerations in Appendix C.

## 4.3 ALTERNATIVE METHODS

**Inference-time methods.** Pre-trained model alignment can be improved at inference by optimizing the latent variables $z_t$ throughout the numerical integration used to generate the (latent) image. This process steers model alignment with an auxiliary loss based on attention maps and fine-grained linguistic analysis of the prompt (e.g., additional input is used to explicitly indicate which words to focus on). In this family, we consider 3 methods: Attend and Excite (A&E) (Chefer et al., 2023b), Structured Diffusion Guidance (SDG) (Feng et al., 2023b) and Semantic-aware Classifier-Free Guidance (SCG) (Shen et al., 2024).

---

[1] We remark that, albeit in a different context, this selection resembles an image retrieval task (Krojer et al., 2023)

[2] We exclude BLIP-VQA for the fine-tuning set composition to avoid biasing the evaluation (Huang et al., 2023).

[3] LoRA adapters (Hu et al., 2021) and fine-tuning also the CLIP-based text encoder do not provide performance improvements (Appendix E.2). Likewise, creating a multi-category model by "merging" different per-category models or using a fine-tuning set composed with images from all categories do not provide performance gains (Appendix E.3).

**Table 1:** Alignment results (%). ▨Gray highlighted▨ style when MI-TUNE outperforms all competitors; Grayed text for under-performing methods per-family; ▨Green heatmaps▨ show per-category absolute gains w.r.t. the base model.

| | Method | BLIP-VQA | | | | | | | HPS | | | | | | | Human (user study) | | | | | | |
|---|---|---|---|---|---|---|---|---|---|---|---|---|---|---|---|---|---|---|---|---|---|---|
| | | Color | Shape | Texture | 2D-Sp. | Non-Sp. | Compl. | (avg) | Color | Shape | Texture | 2D-Sp. | Non-Sp. | Compl. | (avg) | Color | Shape | Texture | 2D-Sp. | Non-Sp. | Compl. | (avg) |
| | SD-2.1-base | 49.65 | 42.71 | 49.99 | 15.77 | 66.23 | 50.53 | (45.81) | 27.64 | 24.56 | 24.99 | 27.50 | 26.66 | 25.70 | (26.17) | 29.76 | 11.90 | 40.48 | 35.71 | 66.67 | 29.76 | (35.71) |
| Infer. | A&E | 61.43 | 47.39 | 64.10 | 16.18 | 66.21 | 51.69 | (51.17) | 28.44 | 24.43 | 25.88 | 28.42 | 26.60 | 25.60 | (26.56) | 31.95 | 15.48 | 52.38 | 32.14 | 65.48 | 30.95 | (38.06) |
| Infer. | SDG | 47.15 | 45.24 | 47.13 | 15.25 | 66.17 | 47.41 | (44.72) | 27.25 | 24.40 | 24.71 | 27.10 | 26.12 | 25.83 | (25.90) | 26.19 | 15.48 | 38.10 | 38.10 | 61.90 | 29.76 | (34.92) |
| Infer. | SCG | 49.82 | 43.28 | 50.16 | 16.31 | 66.60 | 51.07 | (46.21) | 27.86 | 24.85 | 25.57 | 27.76 | 26.98 | 26.03 | (26.51) | 20.24 | 11.90 | 33.33 | 40.48 | 69.05 | 39.29 | (35.71) |
| FT | DPOK | 53.28 | 45.63 | 52.84 | 17.19 | 66.95 | 51.97 | (47.98) | 28.20 | 24.99 | 25.44 | 28.12 | 26.80 | 25.88 | (26.57) | 23.81 | 16.67 | 47.62 | 34.52 | 70.24 | 38.10 | (38.49) |
| FT | GORS | 53.59 | 43.82 | 54.47 | 15.66 | 67.47 | 52.28 | (47.88) | 28.15 | 24.79 | 25.56 | 27.90 | 26.88 | 26.07 | (26.56) | 34.52 | 14.29 | 48.81 | 36.90 | 65.48 | 30.95 | (38.49) |
| FT | HN-ITM | 46.51 | 39.99 | 48.78 | 15.24 | 65.31 | 49.84 | (44.28) | 26.90 | 24.33 | 24.63 | 27.15 | 25.40 | 25.22 | (25.60) | 23.81 | 19.05 | 30.95 | 20.24 | 47.62 | 23.81 | (27.58) |
| | MI-TUNE | 65.04 | 50.08 | 65.82 | †18.51 | 67.77 | 54.17 | (53.56) | 29.13 | 25.57 | 26.20 | †28.50 | 27.15 | 26.70 | (27.21) | 46.43 | 25.01 | 53.19 | †45.24 | 73.81 | 46.43 | (48.35) |
| | *best* Infer.⊟base | 11.78 | 4.68 | 14.11 | 0.54 | 0.37 | 1.16 | (5.44) | 0.80 | 0.29 | 0.89 | 0.92 | 0.32 | 0.33 | (0.59) | 2.19 | 3.58 | 11.90 | 4.77 | 2.38 | 9.53 | (5.72) |
| | *best* FT⊟base | 3.94 | 2.92 | 4.48 | 1.42 | 1.24 | 1.75 | (2.62) | 0.56 | 0.43 | 0.57 | 0.62 | 0.22 | 0.37 | (0.46) | 4.76 | 7.15 | 8.33 | 1.19 | 3.57 | 8.34 | (5.56) |
| | MI-TUNE⊟base | 15.39 | 7.37 | 15.83 | 2.74 | 1.54 | 3.64 | (7.75) | 1.49 | 1.01 | 1.21 | 1.00 | 0.49 | 1.00 | (1.03) | 16.67 | 13.11 | 12.71 | 9.53 | 7.14 | 16.67 | (12.64) |
| | MI-TUNE⊟*best*⋆ | 3.61 | 2.69 | 1.72 | 1.32 | 0.30 | 1.89 | (1.92) | 0.69 | 0.58 | 0.32 | 0.08 | 0.17 | 0.63 | (0.41) | 11.91 | 5.96 | 0.81 | 4.76 | 3.57 | 7.14 | (5.69) |
| | MI-TUNE%*best*⋆ | 5.88 | 5.68 | 2.68 | 7.68 | 0.44 | 3.62 | (4.33) | 2.43 | 2.32 | 1.24 | 0.28 | 0.63 | 2.42 | (1.55) | 34.50 | 31.29 | 1.55 | 11.76 | 5.08 | 18.17 | (17.06) |

A⊟B indicates the absolute difference between A and B; A%B corresponds to the percentage difference (A - B) / B; †: Fine-tuning set obtained from SPRIGHT rather than SD-2.1-base; Human scores do not sum to 100 in each category as users can select multiple methods for each question.

**Fine-tuning methods.** Alternatively, a pre-trained model can be fine-tuned with adapters (Hu et al., 2021) optimized via a variety of RL or supervision methods. Specifically, we consider 3 approaches: Diffusion Policy Optimization with KL regularization (DPOK) (Fan et al., 2023), Generative mOdel finetuning with Reward-driven Sample selection (GORS) (Huang et al., 2023) and Hard-Negatives Image-Text-Matching (HN-ITM) (Krojer et al., 2023). Notice that since results in the literature for both families do not necessarily refer to same base models, to guarantee a fair comparison, we adapted and evaluated all methods on SD-2.1-base.

## 4.4 RESULTS

**Comparing methods.** Table 1 reports the alignment results on T2I-CompBench. To simplify its reading, the bottom part of the table summarizes ($i$) the absolute gain with respect to the SD-2.1-base model for each of the best methods in each family and ($ii$) the percentage gains of MI-TUNE with respect to the alternative method for each category. We also summarize performance as averages across categories for each metric.

Despite performance varies, MI-TUNE achieves a new state of the art across all categories/metrics, often by a sizable margin. While this is more evident for BLIP-VQA and Human, the literature shows that HPS has natural small variations (see Appendix D), hence MI-TUNE gains are significant also for this metric.

Table 1 results are obtained generating fine-tuning sets from SD-2.1-base for all tasks but 2D-Spatial. For this category, we were able to obtain (at best) BLIP-VQA=15.93 and HPS=28.13. Conversely, generating the fine-tuning images from SPRIGHT resulted beneficial. We can link this result to the self-supervision nature of MI-TUNE. On the one hand, our methodology is not bounded to a specific model. On the other hand, the filtering operated via point-wise MI estimation can benefit from "pre-alignment" – MI-TUNE can strengthen existing alignment but might not be sufficient to "induce" it. Notice that all competitors suffer from this trade-off too as no single winner emerges. In particular, despite A&E and GORS are the most frequent best method in their family (winning in 10-out-of-18 scenarios), all competitors show less consistent performance across categories and metrics than MI-TUNE. For instance, for attribute binding (color, shape and texture), fine-tuning methods under-perform according to BLIP-VQA and Human, but the performance gaps are very close considering HPS. Yet, MI-TUNE achieves consistently higher performance across all categories, outperforming alternative fine-tuning methods by a large margin.

Raw alignment performance apart, it is important to highlight MI-TUNE key differences compared to the alternative fine-tuning methods. DPOK uses RL with a reward model (pre-trained with human-labeled real images) to define a prompt-image alignment score to guide the fine-tuning, HN-ITM uses a contrastive

learning approach based on an ad-hoc dataset with real positive (good alignment) and negative (poor alignment) prompt-image pairs, and GORS composes a fine-tuning set generating images from the diffusion model and selecting them based on BLIP-VQA. While GORS is very close in spirit to MI-TUNE, its performance is "biased" – the filtering criteria overlaps with the final evaluation strategy – as explicitly acknowledged by its authors (Huang et al., 2023). Overall, while both DPOK and GORS still require external assistance, MI-TUNE generates images *and* selects them using the target model itself, i.e., it is the first fully self-supervised model for T2I alignment to the best of our knowledge.

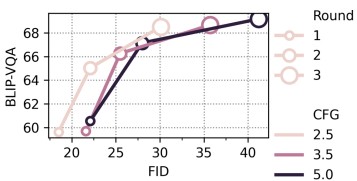

**Figure 2:** Hyper-params search.

**Table 2:** Comparing image quality/variety scores.

| Metric | SD-2.1-base | MI-TUNE ($R=2$, CFG=2.5) | | | | | | | DALLE-3‡ | IMAGEN-3‡ | SDXL‡ |
|---|---|---|---|---|---|---|---|---|---|---|---|
| | | Color | Shape | Texture | Spatial | Non-sp. | Comp. | (avg) | | | |
| FID($\downarrow$) | 17.1 | 22.1 | 16.8 | 17.3 | 18.8 | 16.8 | 20.6 | (18.7) | 20.1 | 17.2 | 13.2 |
| FD-DINO($\downarrow$) | 229.1 | 279.0 | 236.9 | 250.4 | 251.7 | 231.9 | 255.6 | (250.9) | 284.4 | 213.9 | 185.6 |
| CMMD($\downarrow$) | 0.641 | 0.681 | 0.634 | 0.694 | 0.669 | 0.709 | 0.671 | (0.680) | 0.894 | 0.854 | 0.898 |

*Results from 30k samples of* MS-COCO-2014 *validation set;* ‡ *results from (Imagen-Team et al., 2024)*

**Alignment/image quality-variety trade-offs.** MI-TUNE results in Table 1 are obtained from a grid search across multiple fine-tuning rounds $R$ and CFG values. In fact, we observe different trade-offs between alignment and image quality across different configurations. We exemplify this in Figure 2, for the Color category. The figure highlights two opposite dynamics: T2I alignment benefits from multiple fine-tuning rounds (higher BLIP-VQA) but can introduce image artifacts and reduce measured diversity (higher FID). While this trade-off is neither mentioned nor quantified in the literature of the considered methods, it is to be expected – strictly abiding to a prompt impacts the "generative pathways" at sampling time. Interestingly, lowering CFG (typically set to 7.5) counterbalances these dynamics and enables a "sweet spot" – as the model better aligns to a category thanks to fine-tuning, one can alleviate the guidance scale dependency at generation. Table 2 complements this analysis by showing FID, FD-DINO and CMMD scores for all categories, as well for SD-2.1-base and three state of the art models – while all metrics indeed suggest a possible reduction in image variety considering SD-2.1-base, MI-TUNE scores are comparable with other state-of-the-art models (see Figure 3 for example images).

**Table 3:** FT set selection.

| | BLIP-VQA | | HPS | |
|---|---|---|---|---|
| Strategy | Color | Shape | Color | Shape |
| MI only | 65.04 | 50.08 | 29.13 | 25.57 |
| HPS only | 59.43 | 46.87 | n.a. | n.a. |
| MI+Real(0.25) | 61.34 | 48.47 | 29.16 | 25.87 |
| MI+Real(0.5) | 61.63 | 49.50 | 29.38 | 25.92 |
| MI+Real(0.9) | 59.83 | 48.92 | 28.60 | 25.60 |

**Table 4:** Alignment (%) using SDXL.

| Method | BLIP-VQA | | | | | | HPS | | | | | |
|---|---|---|---|---|---|---|---|---|---|---|---|---|
| | Color | Shape | Texture | 2D-Sp. | Non-Sp. | Comp. | Color | Shape | Texture | 2D-Sp. | Non-Sp. | Comp. |
| (ref) SDXL | 60.78 | 49.70 | 55.78 | 21.02 | 68.16 | 52.68 | 28.47 | 24.99 | 25.85 | 28.50 | 26.64 | 25.90 |
| SD-2.1-base | 49.65 | 42.71 | 49.99 | 15.77 | 66.23 | 50.53 | 27.64 | 24.56 | 24.99 | 27.50 | 26.66 | 25.70 |
| MI-TUNE | 69.66 | 55.86 | 66.74 | 22.18 | 72.17 | 57.74 | 29.03 | 25.90 | 27.15 | 29.57 | 27.56 | 26.70 |
| MI-TUNE ⊟ (ref) | 8.88 | 6.16 | 10.96 | 1.16 | 4.01 | 5.06 | 0.56 | 0.91 | 1.30 | 1.07 | 0.92 | 0.80 |
| MI-TUNE % (ref) | 14.61 | 12.39 | 19.65 | 5.52 | 5.88 | 9.61 | 1.97 | 3.64 | 5.03 | 3.75 | 3.45 | 3.09 |

**Table 5:** DiffusionDB.

| Model | HPS |
|---|---|
| SD-2.1-base | 23.99 |
| DiffusionDB | 24.35 |
| MI-TUNE | 25.32 |
| MI-TUNE ⊟ base | 1.33 |
| MI-TUNE ⊟ DiffusionDB | 0.97 |

**Fine-tuning set composition.** The strategy to select prompt-image pairs for the fine-tuning set has a large design space beyond the use of MI. In Table 3, we report (for two categories for brevity) alignment performance using two alternative strategies. Specifically, using HPS rather than MI degrades performance.[4] Results when composing the fine-tuning set by mixing MI-selected and real images selected from the CC2M dataset (Changpinyo et al., 2021) are instead inconsistent (BLIP-VQA steadily degrades but HPS signals an improvement in some scenarios).

**SDXL and DiffusionDB.** We complete our evaluation by presenting results obtained applying MI-TUNE on SDXL in Table 4, and considering an alternative scenario closer to real user application using DiffusionDB in Table 5 to complement the synthetic nature of T2I-CompBench. As expected, "vanilla" SDXL significantly outperforms SD-2.1-base, yet MI-TUNE enables sizable improvements on SDXL alignment (see Figure 4). For the realistic alignment use case in Table 5, we select prompt-images pairs from DiffusionDB and we contrast alignment when fine-tuning using the images already paired with prompts against MI-selected

---

[4]We compute only BLIP-VQA to avoid evaluation bias (Huang et al., 2023).

ones. We use SD-2.1-base as base model and report only HPS scores[5] in Table 5. Overall, fine-tuning with DiffusionDB images improves the base model, yet MI-TUNE enables superior performance (see Figure 5).

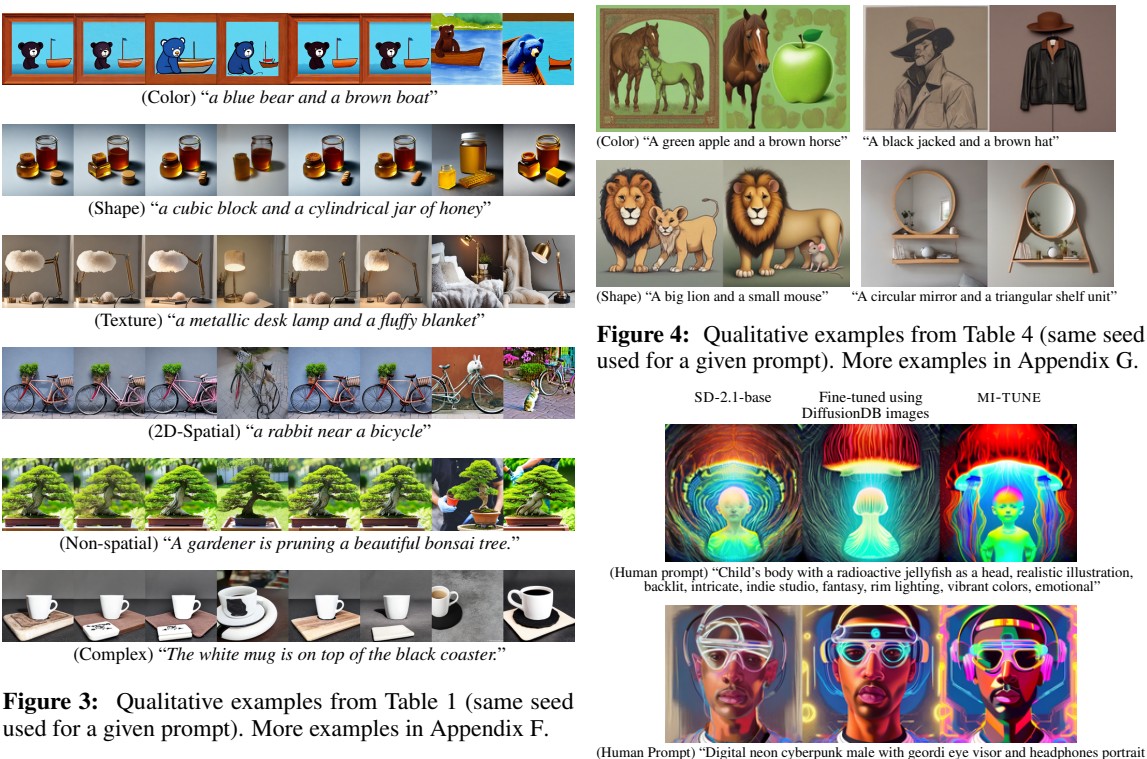

(Color) "*a blue bear and a brown boat*"

(Shape) "*a cubic block and a cylindrical jar of honey*"

(Texture) "*a metallic desk lamp and a fluffy blanket*"

(2D-Spatial) "*a rabbit near a bicycle*"

(Non-spatial) "*A gardener is pruning a beautiful bonsai tree.*"

(Complex) "*The white mug is on top of the black coaster.*"

**Figure 3:** Qualitative examples from Table 1 (same seed used for a given prompt). More examples in Appendix F.

(Color) "A green apple and a brown horse"  "A black jacked and a brown hat"

(Shape) "A big lion and a small mouse"  "A circular mirror and a triangular shelf unit"

**Figure 4:** Qualitative examples from Table 4 (same seed used for a given prompt). More examples in Appendix G.

(Human prompt) "Child's body with a radioactive jellyfish as a head, realistic illustration, backlit, intricate, indie studio, fantasy, rim lighting, vibrant colors, emotional"

(Human Prompt) "Digital neon cyberpunk male with geordi eye visor and headphones portrait painting by donato giancola, kilian eng, john berkey, j. c. leyendecker, alphonse mucha"

**Figure 5:** Qualitative examples from Table 5 (same seed used for a given prompt). More examples in Appendix H.

## 5 CONCLUSION

T2I alignment emerged as an important endeavor to steer image generation to follow the semantics and user intent expressed through a natural text prompt, as it can save considerable manual effort. In this work, we presented a novel approach to improve model alignment, that uses point-wise MI between prompt-image pairs as a meaningful signal to evaluate the amount of information "flowing" between natural text and images. We demonstrated, both qualitatively and quantitatively, that point-wise MI is coherent with existing alignment measures that either use auxiliary VQA models or elicit human intervention.

We presented MI-TUNE, a lightweight, self-supervised fine-tuning method that uses a pre-trained T2I model such as SD to estimate MI, and to generate a synthetic set of aligned prompt-image pairs, which is then used in a parameter-efficient fine-tuning stage, to align the T2I model. Our approach does not require human annotation, auxiliary VQA models, nor costly inference-time techniques, and achieves a new state-of-the-art across all categories/metrics explored in the literature, often by a sizable margin. These results carry on in more complex tasks, and for various base models, illustrating the flexibility of MI-TUNE.

---

[5]The higher prompt complexity does not well suit BLIP-VQA text decomposition (see Appendix H.1).

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

# Appendices

# A DETAILS ON MI ESTIMATION

In this Section, we provide the proof for Equation (2). We start by recalling the definition of the forward and backward processes for a discrete-time diffusion model. For the forward process, we use the following Markov chain

$$q(\boldsymbol{z}_{0:T}, \boldsymbol{p}) = q(\boldsymbol{z}_0, \boldsymbol{p}) \prod_{t=1}^{T} q(\boldsymbol{z}_t \,|\, \boldsymbol{z}_{t-1}), \quad q(\boldsymbol{z}_t \,|\, \boldsymbol{z}_{t-1}) = \mathcal{N}(\boldsymbol{z}_t; \sqrt{1-\beta_t}\boldsymbol{z}_{t-1}, \beta_t I)$$

The backward process (with or without a conditioning signal $\boldsymbol{p}$) evolves according to

$$p_{\boldsymbol{\theta}}(\boldsymbol{z}_{0:T}) = p(\boldsymbol{z}_T) \prod_{t=1}^{T} p_{\boldsymbol{\theta}}(\boldsymbol{z}_{t-1} \,|\, \boldsymbol{z}_t), \quad p_{\boldsymbol{\theta}}(\boldsymbol{z}_{0:T} \,|\, \boldsymbol{p}) = p(\boldsymbol{z}_T) \prod_{t=1}^{T} p_{\boldsymbol{\theta}}(\boldsymbol{z}_{t-1} \,|\, \boldsymbol{z}_t, \boldsymbol{p})$$

where $p_{\boldsymbol{\theta}}(\boldsymbol{z}_{t-1} \,|\, \boldsymbol{z}_t) = \mathcal{N}(\boldsymbol{z}_{t-1}; \mu_{\boldsymbol{\theta}}(\boldsymbol{z}_t), \beta_t I)$, with $\mu_{\boldsymbol{\theta}}(\boldsymbol{z}_t) = \frac{1}{\sqrt{\alpha_t}}\left(\boldsymbol{z}_t - \frac{\beta_t}{\sqrt{1-\bar{\alpha}_t}}\boldsymbol{\epsilon}_{\boldsymbol{\theta}}(\boldsymbol{z}_t, t)\right)$. Similar expressions can be obtained for the conditional version.

Our goal here is to show that the following equality holds

$$\mathbb{E}_{\boldsymbol{p}}[\text{KL}\left[q(\boldsymbol{z}_0 \,|\, \boldsymbol{p}) \,\|\, q(\boldsymbol{z}_0)\right]] = \mathbb{E}_{\boldsymbol{z}, \boldsymbol{p}}[\text{I}(\boldsymbol{z}, \boldsymbol{p})],$$

which is the condition that $\text{I}(\boldsymbol{z}, \boldsymbol{p})$ of Equation (2) should satisfy to be a valid point-wise MI estimator. In particular, we will show that

$$\mathbb{E}_{\boldsymbol{p}}[\text{KL}\left[q(\boldsymbol{z}_0 \,|\, \boldsymbol{p}) \,\|\, q(\boldsymbol{z}_0)\right]] = \mathbb{E}_{t, \boldsymbol{p}, \boldsymbol{z}, \boldsymbol{\epsilon}}\left[\kappa_t \|\boldsymbol{\epsilon}_{\boldsymbol{\theta}}(\boldsymbol{z}_t, \boldsymbol{p}, t) - \boldsymbol{\epsilon}_{\boldsymbol{\theta}}(\boldsymbol{z}_t, \emptyset, t)\|^2\right], \quad \kappa_t = \frac{\beta_t T}{2\alpha_t(1-\bar{\alpha}_t)}.$$

To simplify our proof strategy, we consider the ideal case of perfect training, i.e., $p_{\boldsymbol{\theta}}(\boldsymbol{z}_{0:T}, \boldsymbol{p}) = q(\boldsymbol{z}_{0:T}, \boldsymbol{p})$. Moreover, since $q(\boldsymbol{z}_t \,|\, \boldsymbol{z}_{t-1}, \boldsymbol{p}) = q(\boldsymbol{z}_t \,|\, \boldsymbol{z}_{t-1})$, we can rewrite the $\text{KL}\left[q(\boldsymbol{z}_0 \,|\, \boldsymbol{p}) \,\|\, q(\boldsymbol{z}_0)\right]$ term as follows

$$\text{KL}\left[q(\boldsymbol{z}_0 \,|\, \boldsymbol{p}) \,\|\, q(\boldsymbol{z}_0)\right] = \text{KL}\left[q(\boldsymbol{z}_{0:T} \,|\, \boldsymbol{p}) \,\|\, q(\boldsymbol{z}_{0:T})\right] = \text{KL}\left[p_{\boldsymbol{\theta}}(\boldsymbol{z}_{0:T} \,|\, \boldsymbol{p}) \,\|\, p_{\boldsymbol{\theta}}(\boldsymbol{z}_{0:T})\right] =$$

$$\int p_{\boldsymbol{\theta}}(\boldsymbol{z}_{0:T} \,|\, \boldsymbol{p}) \log \frac{p_{\boldsymbol{\theta}}(\boldsymbol{z}_{0:T} \,|\, \boldsymbol{p})}{p_{\boldsymbol{\theta}}(\boldsymbol{z}_{0:T})} \mathrm{d}\boldsymbol{z}_{0:T} = \int p_{\boldsymbol{\theta}}(\boldsymbol{z}_{0:T} \,|\, \boldsymbol{p}) \sum_{t=1}^{T} \log \frac{p_{\boldsymbol{\theta}}(\boldsymbol{z}_{t-1} \,|\, \boldsymbol{z}_t, \boldsymbol{p})}{p_{\boldsymbol{\theta}}(\boldsymbol{z}_{t-1} \,|\, \boldsymbol{z}_t)} \mathrm{d}\boldsymbol{z}_{0:T} =$$

$$\sum_{t=1}^{T} \int p_{\boldsymbol{\theta}}(\boldsymbol{z}_{0:t-2, t:T} \,|\, \boldsymbol{p}) \left(\int p_{\boldsymbol{\theta}}(\boldsymbol{z}_{t-1} \,|\, \boldsymbol{z}_t, \boldsymbol{p}) \log \frac{p_{\boldsymbol{\theta}}(\boldsymbol{z}_{t-1} \,|\, \boldsymbol{z}_t, \boldsymbol{p})}{p_{\boldsymbol{\theta}}(\boldsymbol{z}_{t-1} \,|\, \boldsymbol{z}_t)} \mathrm{d}\boldsymbol{z}_{t-1}\right) \mathrm{d}\boldsymbol{z}_{0:t-2, t:T} =$$

$$\sum_{t=1}^{T} \int p_{\boldsymbol{\theta}}(\boldsymbol{z}_t \,|\, \boldsymbol{p}) \text{KL}\left[p_{\boldsymbol{\theta}}(\boldsymbol{z}_{t-1} \,|\, \boldsymbol{z}_t, \boldsymbol{p}) \,\|\, p_{\boldsymbol{\theta}}(\boldsymbol{z}_{t-1} \,|\, \boldsymbol{z}_t)\right] \mathrm{d}\boldsymbol{z}_t =$$

$$\sum_{t=1}^{T} \frac{1}{2\beta_t} \int p_{\boldsymbol{\theta}}(\boldsymbol{z}_t \,|\, \boldsymbol{p}) \|\mu_{\boldsymbol{\theta}}(\boldsymbol{z}_t) - \mu_{\boldsymbol{\theta}}(\boldsymbol{z}_t, \boldsymbol{p})\|^2 \mathrm{d}\boldsymbol{z}_t =$$

$$\sum_{t=1}^{T} \frac{1}{2\beta_t} \frac{\beta_t^2}{\alpha_t(1-\bar{\alpha}_t)} \int p_{\boldsymbol{\theta}}(\boldsymbol{z}_t \,|\, \boldsymbol{p}) \|\boldsymbol{\epsilon}_{\boldsymbol{\theta}}(\boldsymbol{z}_t) - \boldsymbol{\epsilon}_{\boldsymbol{\theta}}(\boldsymbol{z}_t, \boldsymbol{p})\|^2 \mathrm{d}\boldsymbol{z}_t =$$

$$\mathbb{E}_{t, \boldsymbol{z}_t}\left[\kappa_t \|\boldsymbol{\epsilon}_{\boldsymbol{\theta}}(\boldsymbol{z}_t, \emptyset, t) - \boldsymbol{\epsilon}_{\boldsymbol{\theta}}(\boldsymbol{z}_t, \boldsymbol{p}, t)\|^2\right] =$$

$$\mathbb{E}_{t, \boldsymbol{z}, \boldsymbol{\epsilon}}\left[\kappa_t \|\boldsymbol{\epsilon}_{\boldsymbol{\theta}}(\boldsymbol{z}_t, \emptyset, t) - \boldsymbol{\epsilon}_{\boldsymbol{\theta}}(\boldsymbol{z}_t, \boldsymbol{p}, t)\|^2\right], \quad \kappa_t = \frac{\beta_t T}{2\alpha_t(1-\bar{\alpha}_t)}$$

which allows to prove that the quantity in Equation (2) is indeed a valid point-wise MI estimator.

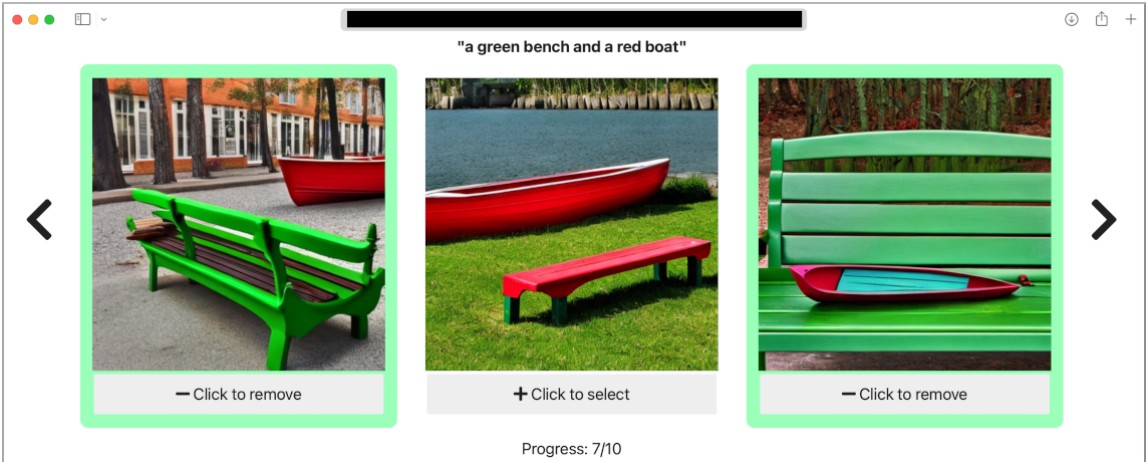

**Figure 6:** Web app screenshot example of the alignment metric comparison survey.

## B   DETAILS ON USER STUDY

Our user studies are based on small focus groups participants only, with (lightly) guided discussions led by a moderator. In those campaigns we elicited feedback from users regarding the comparison of different alignment metrics, aiming to understand if MI is a *plausible* choice. Although launching large-scale survey campaigns would be desirable, this would require a completely different organization and implementation with respect to what what we adopted for this work.

**The survey web app.** Beside *punctually* comparing alignment metrics § 3 and methods § 4.4, we designed a web app to collect *subjective* feedback, in the form of mini surveys, from real users. Each survey is composed of multiple tests, each showing a prompt and a set of images generated from it. Under the hood, the web app corresponds to a jupyter notebook with `ipywidgets`[6] for UI controls, rendered via the `voila`[7] framework and deployed live via a docker-ized HuggingFace space. Via the web app we run campaigns to *compare alignment metrics* and to *compare alignment methods*.

Figure 6 shows an example screenshot of the alignment metric comparison survey § 3. As from the example, users are free to select from 0 to up to 3 images for each prompt. However, to stress users subjectivity, we intentionally did not provide guidelines on how to handle "odd" cases (e.g., if the prompt asks for a picture of "an apple", but the picture show more than one apple). Last, each survey is saved as a separate CSV with the timestamp of its creation which also serves as unique identifier of the survey, i.e., neither a user identifier nor cookies are required by the web app logic, so users' privacy and anonymity is preserved.

### B.1   COMPARING ALIGNMENT METRICS

In the first surveys campaign we aimed to understand how users perceive images pre-selected by BLIP-VQA, HPS and MI. Specifically, we run surveys composed of 10 tests, each showing a prompt and the related best image among 50 generations (using SD-2.1-base) as ranked according to each metric separately (§ 3). Each of the 10 prompts is randomly selected from a pool of 700 prompts for the T2I-Combench color category, and at each test the order in which the 3 pictures is shown is also randomized.

---

[6]https://ipywidgets.readthedocs.io/en/stable/
[7]https://voila.readthedocs.io/en/stable/using.html

**Table 6:** User study about comparing alignment metrics.

| Metric | | | Campaign answers (%) | | | |
|---|---|---|---|---|---|---|
| MI | BLIP-VQA | HPS | Academic users | Random users | Students | *avg* |
| ○ | ○ | ○ | 14.7 | 16.9 | 25.0 | *18.9* |
| ○ | ○ | ● | 1.8 | 14.0 | 2.7 | *6.2* |
| ○ | ● | ○ | 10.4 | 22.0 | 4.1 | *12.2* |
| ○ | ● | ● | 4.0 | 7.4 | 3.6 | *5.0* |
| ● | ○ | ○ | 4.0 | 10.6 | 0.9 | *5.2* |
| ● | ○ | ● | 6.0 | 2.5 | 2.3 | *3.6* |
| ● | ● | ○ | 18.7 | 10.6 | 16.8 | *15.4* |
| ● | ● | ● | 40.4 | 16.0 | 44.6 | *33.7* |
| ● | ◑ | ◑ | 69.1 | 39.7 | 64.6 | *57.8* |
| ◑ | ● | ◑ | 73.5 | 56.0 | 69.1 | *66.2* |
| ◑ | ◑ | ● | 52.2 | 39.9 | 53.2 | *48.2* |

●(selected)  ○(not selected)  ◑(indifferent to the selection)

We run surveys across three user groups: *Academic users* (5 members) are representative of highly informed and tech savvy users, who are familiar with how generative models work; *Random users* (25 members) are representative of illiterate users who are not familiar with computer-based image generation; *Students* (16 members) are representative of masters' level students who are familiar with image generation tools, and who have attended introductory-level machine learning classes.

Overall, we collected 102 surveys (45, 35 and 22 surveys across 3 days for Academics, Random users and Students respectively) which we detail in Table 6. The top part of the table breaks down all possible answers combinations. The results, although with some differences between user groups, clearly highlight that the three alignment metrics we consider in this work are roughly equivalent, with MI and BLIP-VQA being preferred over HPS. For the Academics and Students groups, all the three images are considered sufficiently aligned with the prompt in almost half of the cases ($40.4\%$ and $44.6\%$ respectively). Interestingly, random users select only one of the three images about $10\times$ much more frequently than the other two groups (on average 14.2% for real users while 5.4% and 3% for Academics and Students respectively). We hypothesise that being previously exposed (or not) to the technical problems of image generation from the alignment perspective, or simply being literate (or not) about machine learning can influence the selection among the three pictures.

The bottom part of the table summarizes the answers for each individual metric. Despite the general preference for BLIP-VQA, the results corroborate once more that MI provides a meaningful alignment signal (possibly compatible with aesthetics too).

Finally, we recall that our goal in this section is to study whether **MI is a *plausible* alignment measure**, rather than electing the "best" alignment metric. Indeed, this analysis does not indicate the final performance of alignment methods, which instead we report in Table 1.

## B.2 COMPARING ALIGNMENT METHODS

In this second survey campaign we aimed to understand how users perceive images generated by the 8 methods we considered in our study, i.e., "vanilla" SD, A&E (Chefer et al., 2023b), SDG (Feng et al., 2023b) and SCG (Salimans & Ho, 2022) DPOK (Fan et al., 2023), GORS (Huang et al., 2023), HN-ITM (Krojer et al., 2023) and our method Mutual Information Fine Tuning (MI-TUNE) (when used with a single round of fine-tuning).

To do so, we run surveys composed of 2 tests for each T2I-Combench category (12 rounds in total). Each test shows a prompt and the 8 pictures generated using a different method. For each category, we randomly

**Table 7:** Users study comparing alignment methods. **Bold** shows best performance; ⋄shows the best method per-family.

| Alignment Methodology | Model | Category (%) | | | | | |
|---|---|---|---|---|---|---|---|
| | | Color | Shape | Texture | 2D-Spatial | Non-spatial | Complex |
| *none* | SD-2.1-base | 29.76 | 11.90 | 40.48 | 35.71 | 66.67 | 29.76 |
| Inference-time | A&E | ⋄31.95 | ⋄15.48 | ⋄52.38 | 32.14 | 65.48 | 30.95 |
| | SDG | 26.19 | ⋄15.48 | 38.10 | 38.10 | 61.90 | 29.76 |
| | SCG | 20.24 | 11.90 | 33.33 | ⋄40.48 | ⋄69.05 | ⋄39.29 |
| Fine-tuning | DPOK | 23.81 | 16.67 | ⋄47.62 | 34.52 | ⋄70.24 | ⋄38.10 |
| | GORS | ⋄34.52 | 14.29 | 48.81 | ⋄36.90 | 65.48 | 30.95 |
| | HN-ITM | 23.81 | ⋄19.05 | 30.95 | 20.24 | 47.62 | 23.81 |
| | MI-TUNE | **46.43** | **25.01** | **53.19** | **45.24** | **73.81** | **46.43** |

selected 100 prompts from T2I-Combench test set to pre-generate the pictures. At run time, the web app randomly selects 2 prompts for each category, and also randomly selects images from the related pool. Last, it randomly arranges both the tests (so that categories are shuffled) and the methods (so that pictures of a method are not visualized in the same position in the visualized grid).

Table 7 collects the results of a campaign with 42 surveys. Specifically, the table shows the percentage of answers where the picture of a given method was selected (no matter if other methods were also selected) – theses results are integrated in right side of Table 1 and are duplicated here for completeness.

## C EXPERIMENTAL PROTOCOL DETAILS

We report in Table 8 all the hyper-parameters we used for our experiments.

**Table 8:** Training hyperparameters.

| Name | Value |
|---|---|
| Trainable model | UNET |
| Trainable timesteps | $t \sim U(500, 1000)$ |
| PEFT | DoRA (Liu et al., 2024) |
| Rank | 32 |
| $\alpha$ | 32 |
| Learning rate (LR) | $1e - 4$ |
| Gradient norm clipping | 1.0 |
| LR scheduler | Constant |
| LR warmup steps | 0 |
| Optimizer | AdamW |
| AdamW - $\beta_1$ | 0.9 |
| AdamW - $\beta_2$ | 0.999 |
| AdamW - weight decay | $1e - 2$ |
| AdamW - $\epsilon$ | $1e - 8$ |
| Resolution | $512 \times 512$ |
| Classifier-free guidance scale | 7.5 |
| Denoising steps | 50 |
| Batch size | 400 |
| Training iterations | 300 |
| GPUs for Training | $1 \times$ NVIDIA A100 |

Next, we provide additional details on the computational cost of MI-TUNE. In our approach, there are two distinct phases that require computational effort:

**Generation**: The first is the construction of the fine-tuning set $\mathcal{S}$ based on point-wise MI. As a reminder, for this phase, we use a pre-trained SD model (namely SD-2.1-base at a resolution $512 \times 512$) and, given a prompt, conditionally generate 50 images, while at the same time computing point-wise MI between the prompt and each image. This is done for all the prompts in the set $\mathcal{P}$. Specifically for T2I-Combench, each category training set has 700 prompts, and for each prompt we generate 50 images from which we select the one with highest MI. The generation of the $700 \times 50$ fine-tuning set requires roughly *24 hours, i.e., about 2min per-prompt* on a single A100-80GB GPU – the 50 images are generated together (as they roughly require 50GB of the 80GB available VRAM), while each prompt is processed sequentially.

**Fine-Tuning**: The second is the parameter efficient fine-tuning of the pre-trained model. Using the configuration discussed above, MI-TUNE requires *8 hours* when using a single A100-80GB GPU.

Note that ($i$) there is no overhead at image generation time: once a pre-trained model has been fine-tuned with MI-TUNE, conditional sampling takes the same amount of time of "vanilla" SD and ($ii$) while we report computational costs considering a single GPU, this is a extreme scenario and the time to process the workloads scales down (almost linearly) with the number of GPUs used according to our observations.

## D HPS SCORES RANGE

Wu et al. (2023a) report a detailed benchmark of their metrics across 20+ models in the HPS-v2 GitHub repository `https://github.com/tgxs002/HPSv2`. These details are hidden by default when loading the repository home page and need to be explicitly "opened" expanding collapsed menus (e.g., ▶ v2 benchmark). To ease discussion, in Table 9 we report an extract of these benchmarks focusing on StableDiffusion as other models are out scope for our study.

**Table 9:** HPS benchmark across multiple Stable Diffusion models extracted for HPS-v2 GitHub repo.

| Benchmark | Model | Animation | Concept-Art | Painting | Photo | (*avg*) |
|---|---|---|---|---|---|---|
| v2 | SDXL Refiner (0.9) | 28.45 | 27.66 | 27.67 | 27.46 | (*27.80*) |
| | SDXL Base (0.9) | 28.42 | 27.63 | 27.60 | 27.29 | (*27.73*) |
| | SD (2.0) | 27.48 | 26.89 | 26.86 | 27.46 | (*27.17*) |
| | SD (1.4) | 27.26 | 26.61 | 26.66 | 27.27 | (*26.95*) |
| v2.1 | SDXL Refiner (0.9) | 33.26 | 32.07 | 31.63 | 28.38 | (*31.34*) |
| | SDXL Base (0.9) | 32.84 | 31.36 | 30.86 | 27.48 | (*30.63*) |
| | SD (2.0) | 27.09 | 26.02 | 25.68 | 26.73 | (*26.38*) |
| | SD (1.4) | 26.03 | 24.87 | 24.80 | 25.70 | (*25.35*) |

Results refer to two benchmark and are visually split between SD and SDXL. The columns Animation, Concept-Art, Painting and Photo are different images style, while (*avg*) reflects average by row.

Both versions of the benchmark present similar takeaways which we can summarize in two main observations. Specifically, ($i$) different versions of the same model present $< 0.5$ differences and ($ii$) SDXL outperforms SD of about +1 point – the variation of HPS scores is extremely contained even if these models are different generations apart.

Our HPS scores in Table 1 present similar properties, but other literature (e.g., Table 2 in Zhao et al. (2024)) present similar evidence.

# E  ADDITIONAL RESULTS AND ABLATIONS

## E.1  ABLATION: FINE-TUNING SET SELECTION STRATEGIES

**Fine-tuning set selection strategy.** It is important to stress that creating a fine-tuning dataset using the very same metric used for the final evaluation can artificially introduce a bias as stated in (Huang et al., 2023): "calculating the rewards for GORS with the automatic evaluation metrics can lead to biased results".

The selection strategy to compose the fine-tuning dataset is directly related to alignment scores and different fine-tuning methods opt for different choices. Specifically: HN-ITM uses an ad-hoc dataset with real positive and negative pairs; GORS uses a synthetic dataset with no selection, but the fine-tuning loss of each sample is weighted by BLIP-VQA DPOK synthesizes new images at each training iteration since it is an online RL fine-tuning approach, and uses a pre-trained human preference model for reward. Table 1, in the main paper, shows alternative fine-tuning strategies based on synthetic generated data using a variety of selection scores: GORS and DPOK are the closest methods to MI-TUNE from this point of view, yet generally underperforming compared to it.

**Table 10:** FT set selection.

| Strategy | BLIP-VQA | | HPS | |
|---|---|---|---|---|
| | Color | Shape | Color | Shape |
| MI only | 65.04 | 50.08 | 29.13 | 25.57 |
| HPS only | 59.43 | 46.87 | *n.a.* | *n.a.* |
| MI+Real(0.25) | 61.34 | 48.47 | 29.16 | 25.87 |
| MI+Real(0.5) | 61.63 | 49.50 | 29.38 | 25.92 |
| MI+Real(0.9) | 59.83 | 48.92 | 28.60 | 25.60 |

For completeness, we perform an experiment where we fine-tune based on a dataset selected via HPS scores. Results in Table 10 (same as Table 3, but duplicated here for simplicity) show that selecting fine-tuning samples based on MI outperforms such an alternative strategy, using BLIP-VQA.

Next, another natural question to ask is whether the self-supervised fine-tuning method we suggest in this work is a valid strategy. Indeed, instead of using synthetic image data for fine-tuning the base model, it is also possible to use real-life, captioned image data. Then, we present an ablation on the use of real samples, along with synthetic images, in the fine-tuning procedure. In Table 10(bottom) we report the experimental results obtained by composing the fine-tuning dataset by imposing the ratio of images generated by the SD model to $x$, and the ratio of real images taken from the CC12M dataset (Changpinyo et al., 2021) to $(1 - x)$, where in both cases we select the candidate images to be used in the fine-tuning set $\mathcal{S}$ using MI. So, for example, MI+Real(0.25) indicates that we use 25% of real images. Interestingly, we observe the following trend. Complementing the synthetically generated samples with few real ones does not benefit alignment (lower BLIP-VQA) but might have a positive effect for aesthetics (higher HPS).

**Fine-tuning set size.** We continue by reporting an ablation on the fine-tuning set $\mathcal{S}$ size.

Specifically, based on Algorithm 1, two parameters determine both the quality and the associated computational cost related to the fine-tuning set $\mathcal{S}$: the number of candidate images $M$, and how many $k$ are selected to be included in $\mathcal{S}$.

**Table 11:** BLIP-VQA alignment results on T2I-CompBench's Color and Shape categories varying size and composition of fine-tuning set. Results obtained using $R$=1.

| Hyper-params | | Category | |
|---|---|---|---|
| $M$ | $k$ | Color | Shape |
| 30 | 1 | 58.12 | 47.48 |
| 50 | 7 | 59.31 | 47.26 |
| 50 | 1 | 61.57 | 48.40 |
| 100 | 1 | 60.12 | 47.80 |
| 500 | 1 | 59.28 | 46.79 |

Table 11 shows that the best performance is obtained selecting $2\%$ images (1 image out of 50). We repeated the finetuning experiments on the categories Color and Shape by varying the selection ratio in the ranges $\{7/50, 1/30, 1/100, 1/500\}$. Results indicate that the best selection ratio is the middle-range corresponding to the baseline MI-TUNE. We hypothesise that higher selection ratios pollute the fine-tuning set with lower quality images, while a more selective threshold favours images which have the highest alignment but possibly lower realism. Additionally, we remark that the number $M$ of candidate images has a negligible impact, above $M = 50$, whereas fewer candidate images induce degraded performance. Hence, the value $M = 50$ is, in our experiments, a sweet-spot that produces a valid candidate set, while not imposing a large computational burden.

## E.2   ABLATION: FINE-TUNING MODEL ADAPTERS AND MODALITIES

In this Section, we provide additional results (Table 12) on MI-TUNE, concerning which part of the pre-trained SD model to fine-tune. In particular, we tried to fine-tune the denoising UNET network alone and both the denoising and the text encoding (CLIP) networks. The baseline results are obtained, as described in the main paper, with Do-RA (Liu et al., 2024) adapters. Switching to Lo-RA layers Hu et al. (2021) incurs in a performance degradation, a trend observed also for other tasks in the literature (Liu et al., 2024). Interestingly, joint fine-tuning of the UNET backbone together with the text encoder layers degrades performance as well, which has also been observed in the literature Huang et al. (2023). Even if, in principle, a joint fine-tuning strategy should provide better results, as the amount of information transferred from the prompt to the image is bottle-necked by the text encoder architecture, we observed empirically more unstable training dynamics than the variant where only the score network backbone is fine-tuned, resulting in degraded performance.

**Table 12:** BLIP-VQA alignment results on T2I-CompBench's Color and Shape categories finetuning different portions of the model.

| Model | Category | |
|---|---|---|
| | Color | Shape |
| MI-TUNE DoRA | 61.57 | 48.40 |
| MI-TUNE LoRA | 58.25 | 48.27 |
| MI-TUNE UNet+Text(joint) | 57.88 | 47.79 |

## E.3   ABLATION: COMBINING CATEGORIES INTO A SINGLE MODEL

The design space for T2I alignment improvement has many options and this should call not only to investigate alignment performance but also operational and computational costs. For instance, fine-tuning methods require to create ad-hoc models while one can argue that a single/multi-purpose model might be a more lean and general solution.

This calls for investigating if/how different task-specific fine-tuned models can be combined into a single model to address the different tasks at once. For the T2I-Combench, we considered two design options:

1. **Weights merging**: the DoRA weights of the 6 distinct per-category models are "merged" doing their arithmethic means forming a new "meta" model.

2. **Joint optimization**: we create a new "meta" model by running a single fine-tuning process but using the union of the category-specific fine-tuning set.

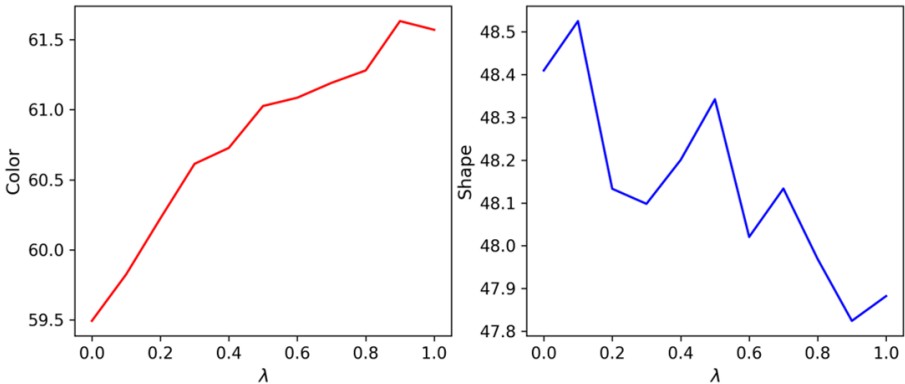

**Figure 7:** Weights merging: $\lambda \times$ Color $+ (1.0 - \lambda) \times$ Shape.

To start from a reference example, Fig. 7 reports the BLIP-VQA obtained when testing on the color and shape test sets on the merged model obtained of the two task-specific models. The hyper-parameter $\lambda$ is used to balance the merging. For instance, at $\lambda = 0$, the performance on color (left plot) are obtained using the shape-only model. Overall, the results show that these two categories are (partially) conflicting across all $\lambda$ values. Yet, a performance trade off might be sufficient in some scenarios.

**Table 13:** Benchmarking strategies for combining models.

| MI-TUNE *variants* | Color (BLIP-VQA) | Shape (BLIP-VQA) | Texture (BLIP-VQA) | 2D-Spatial (UNIDET) | Non-spatial (BLIP-VQA) | Complex (BLIP-VQA) |
|---|---|---|---|---|---|---|
| *from* Table 1 | **61.57** | **48.40** | **58.27** | **18.51** | 67.77 | 53.54 |
| Model weighting | 58.50 | 48.23 | 58.22 | 16.72 | 68.28 | 54.35 |
| Joint optimization | 60.35 | 47.73 | 57.96 | 18.44 | **69.68** | **54.88** |

We then extended the analysis across all categories using a simple arithmetic mean for model merging, i.e., all models have the same weight. Results are reported in Table 13 using MI-TUNE as reference. Overall, for most categories, the single "meta" model has degraded performance and neither weights merging nor joint optimization are the best alternative across all categories.

# F  QUALITATIVE EXAMPLES FOR T2I-COMPBENCH USING SD-2.1-BASE

## F.1  COLOR PROMPTS

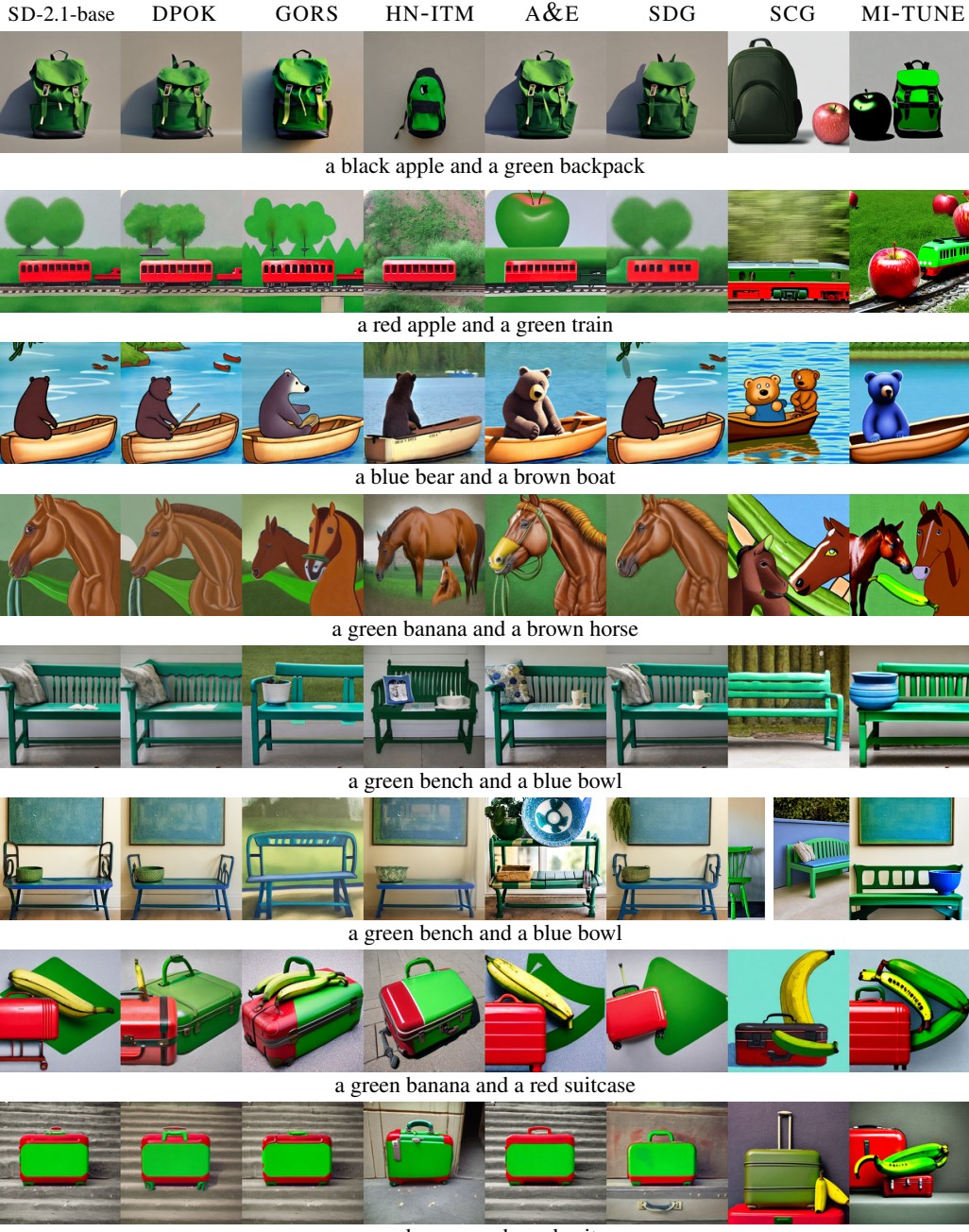

**Figure 8:** Qualitative examples for Color from Table 1 (same seed used for a given prompt).

## F.2 SHAPE PROMPTS

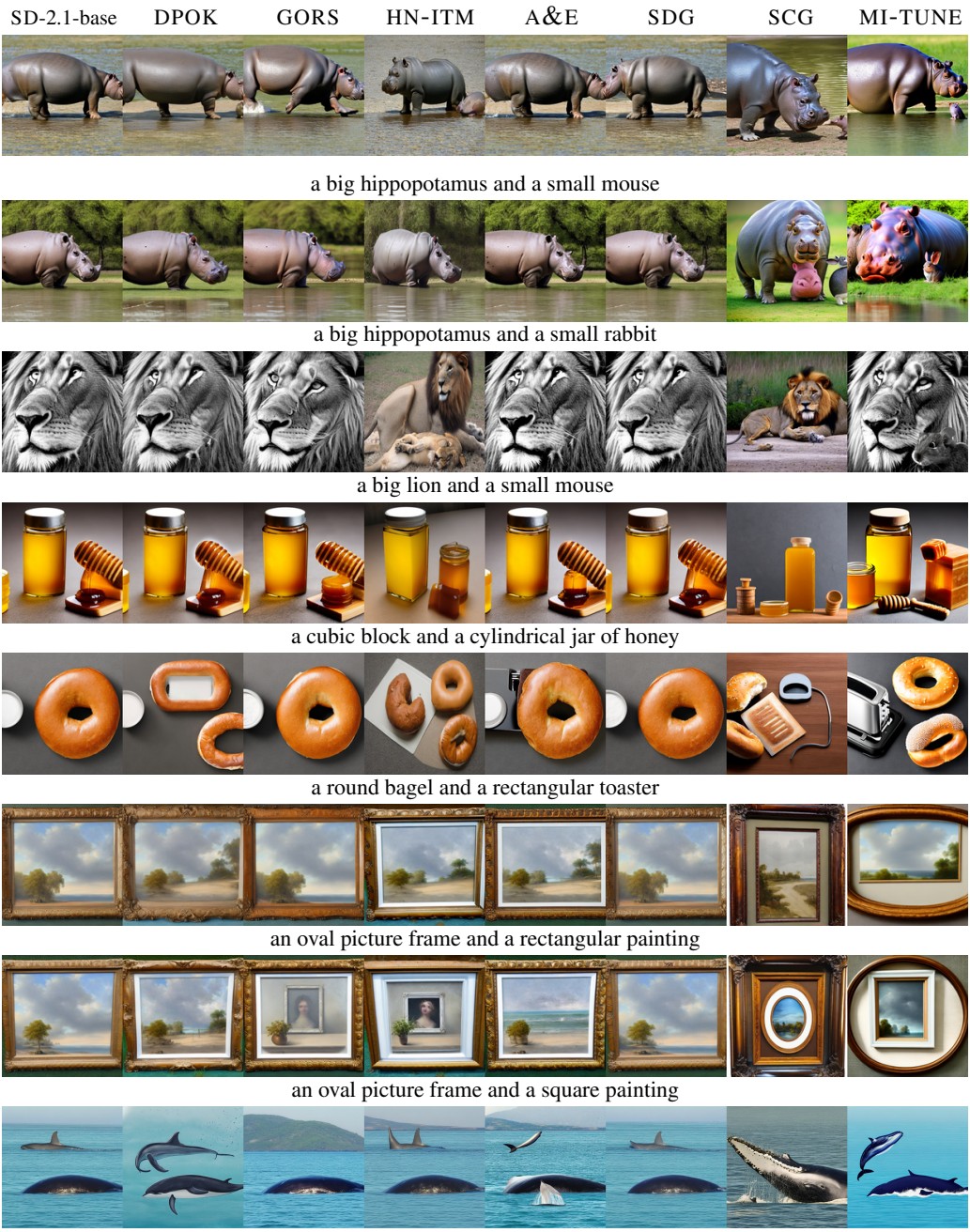

**Figure 9:** Qualitative examples of the Shape category from Table 1 (same seed used for a given prompt).

## F.3 TEXTURE PROMPTS

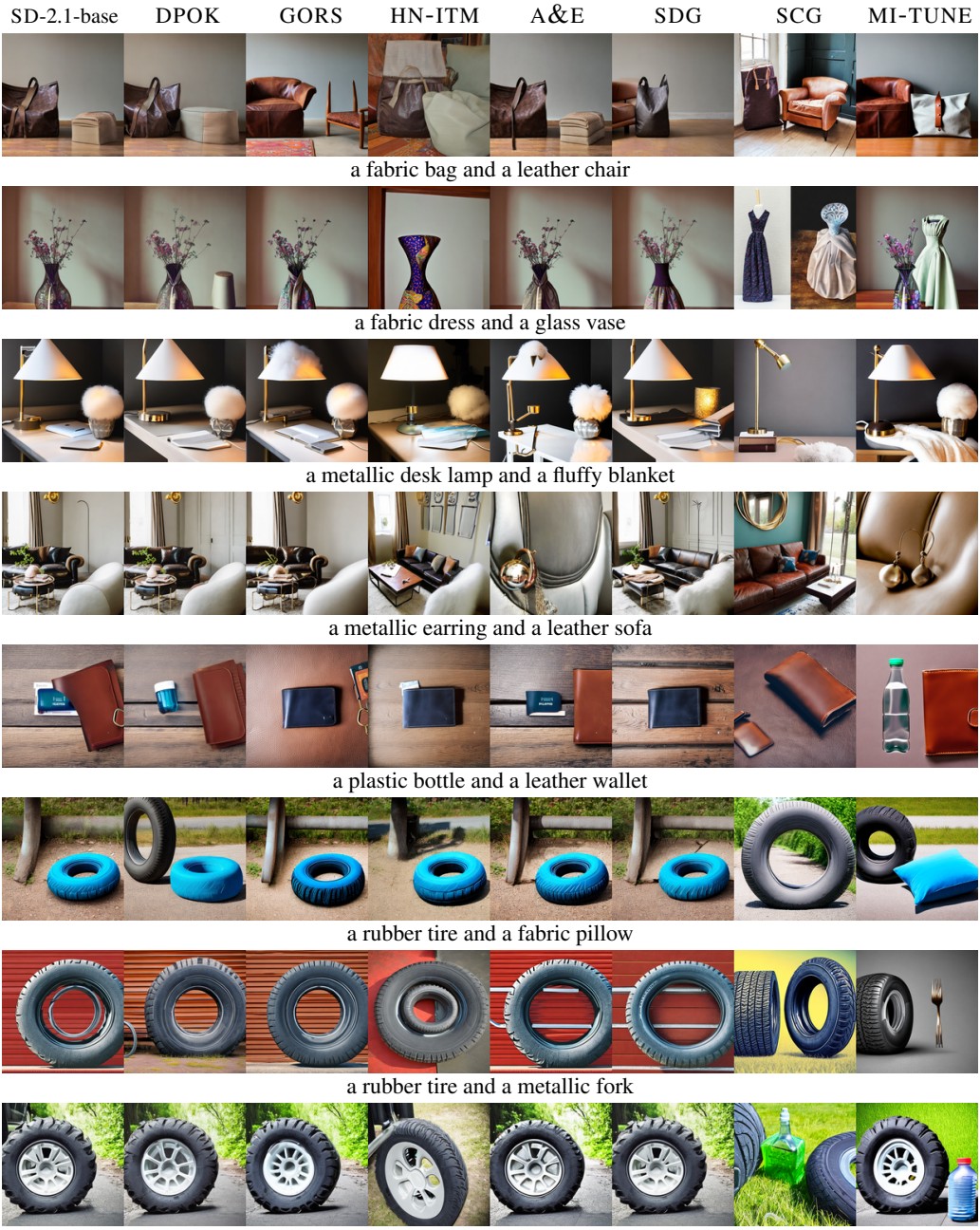

**Figure 10:** Qualitative examples of the Texture category from Table 1 (same seed used for a given prompt).

## F.4    2D-SPATIAL PROMPTS

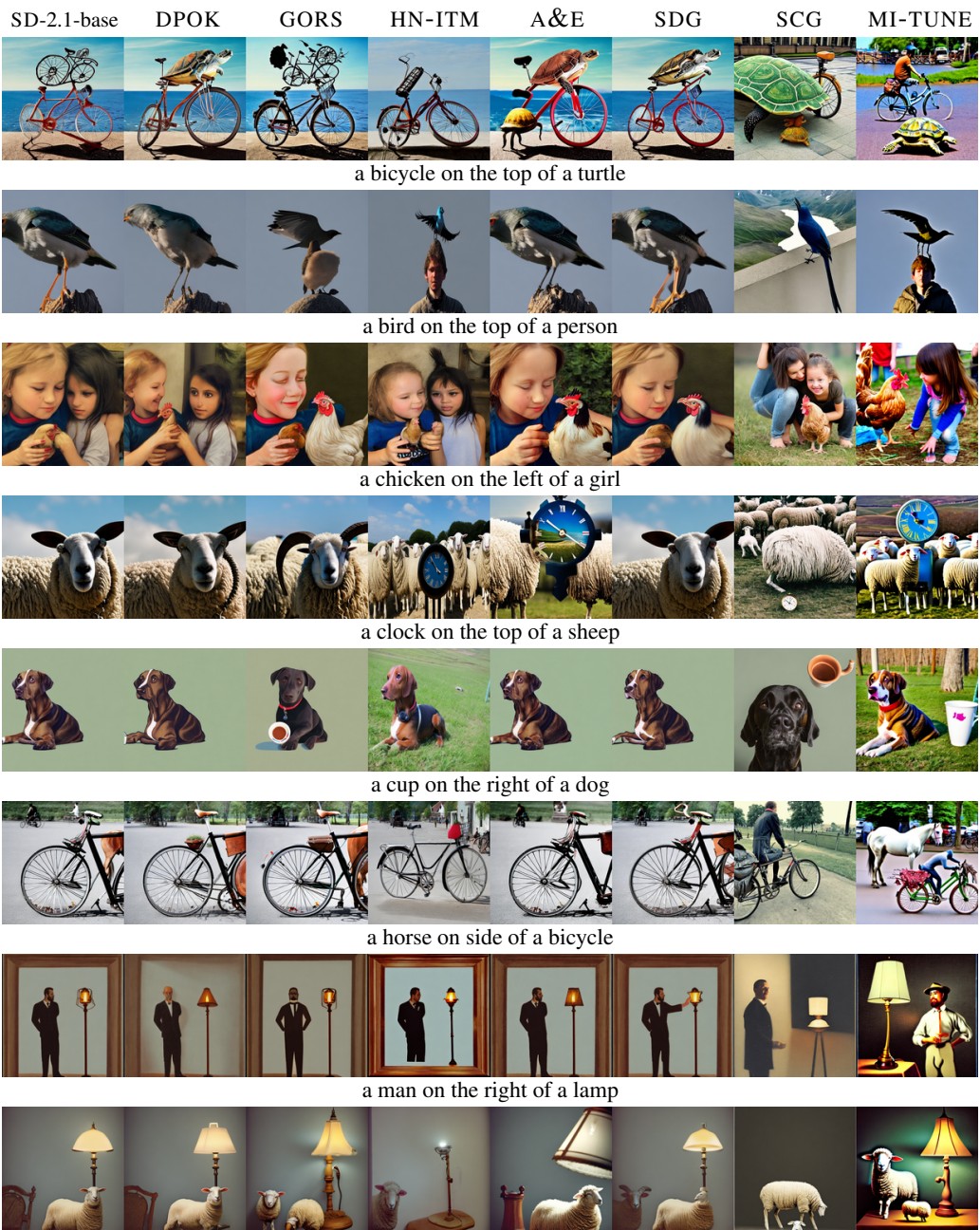

Figure 11: Qualitative examples of the 2D-Spatial category from Table 1 (same seed used for a given prompt).

## F.5 NON-SPATIAL PROMPTS

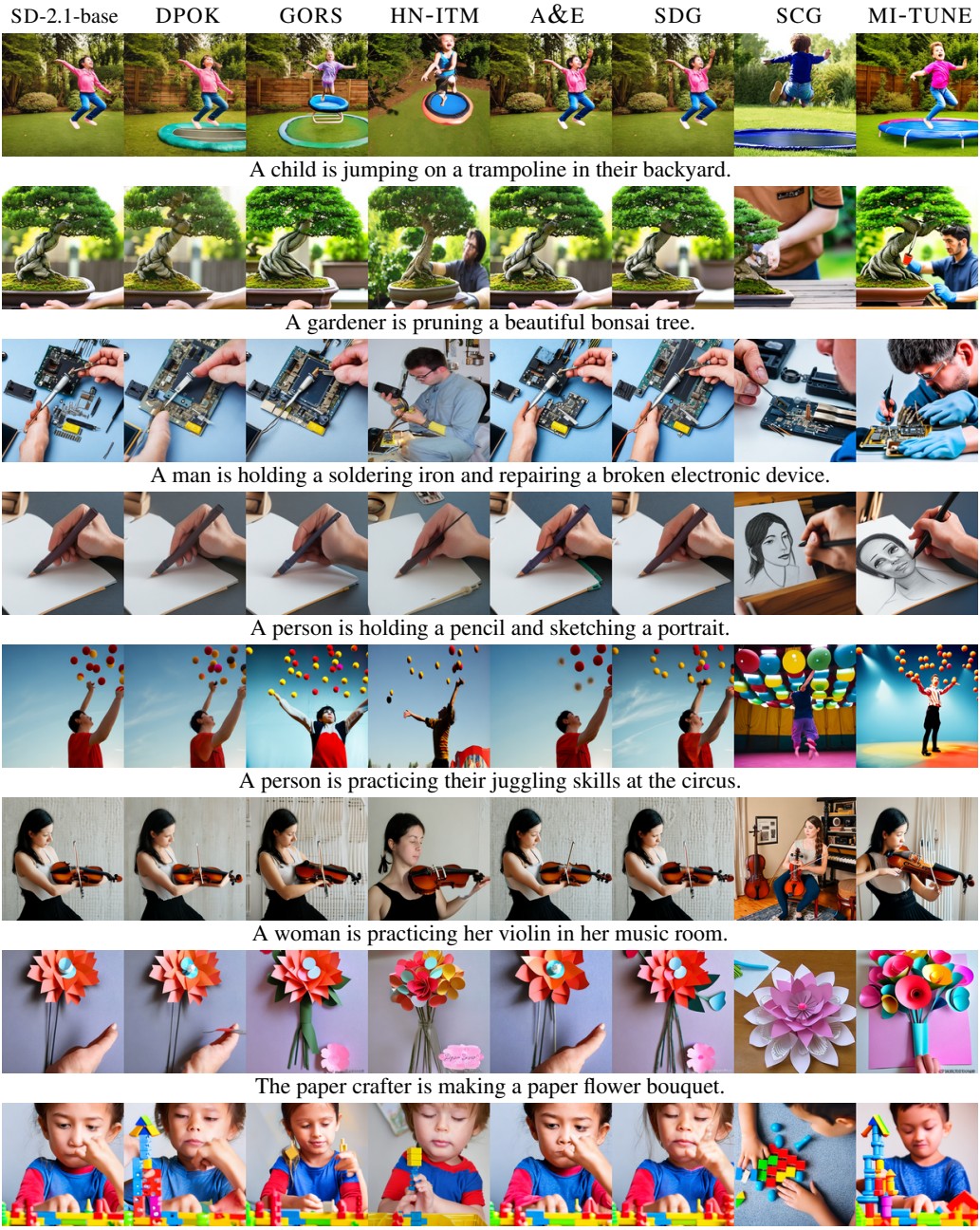

**Figure 12:** Qualitative examples of the Non-spatial category from Table 1 (same seed used for a given prompt).

## F.6 COMPLEX PROMPTS

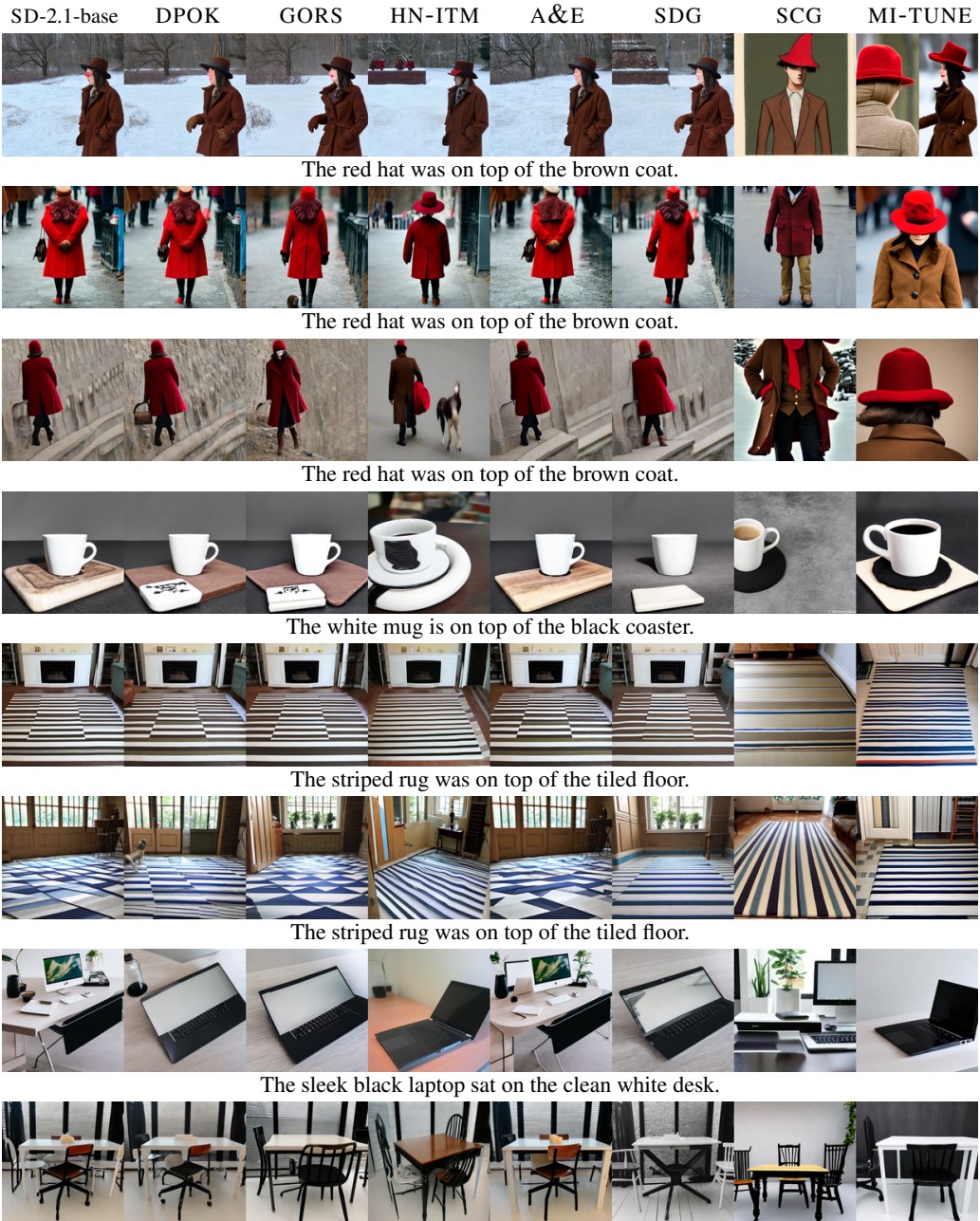

**Figure 13:** Qualitative examples of the Complex category from Table 1 (same seed used for a given prompt).

## G  QUALITATIVE EXAMPLES FOR T2I-COMPBENCH USING SDXL

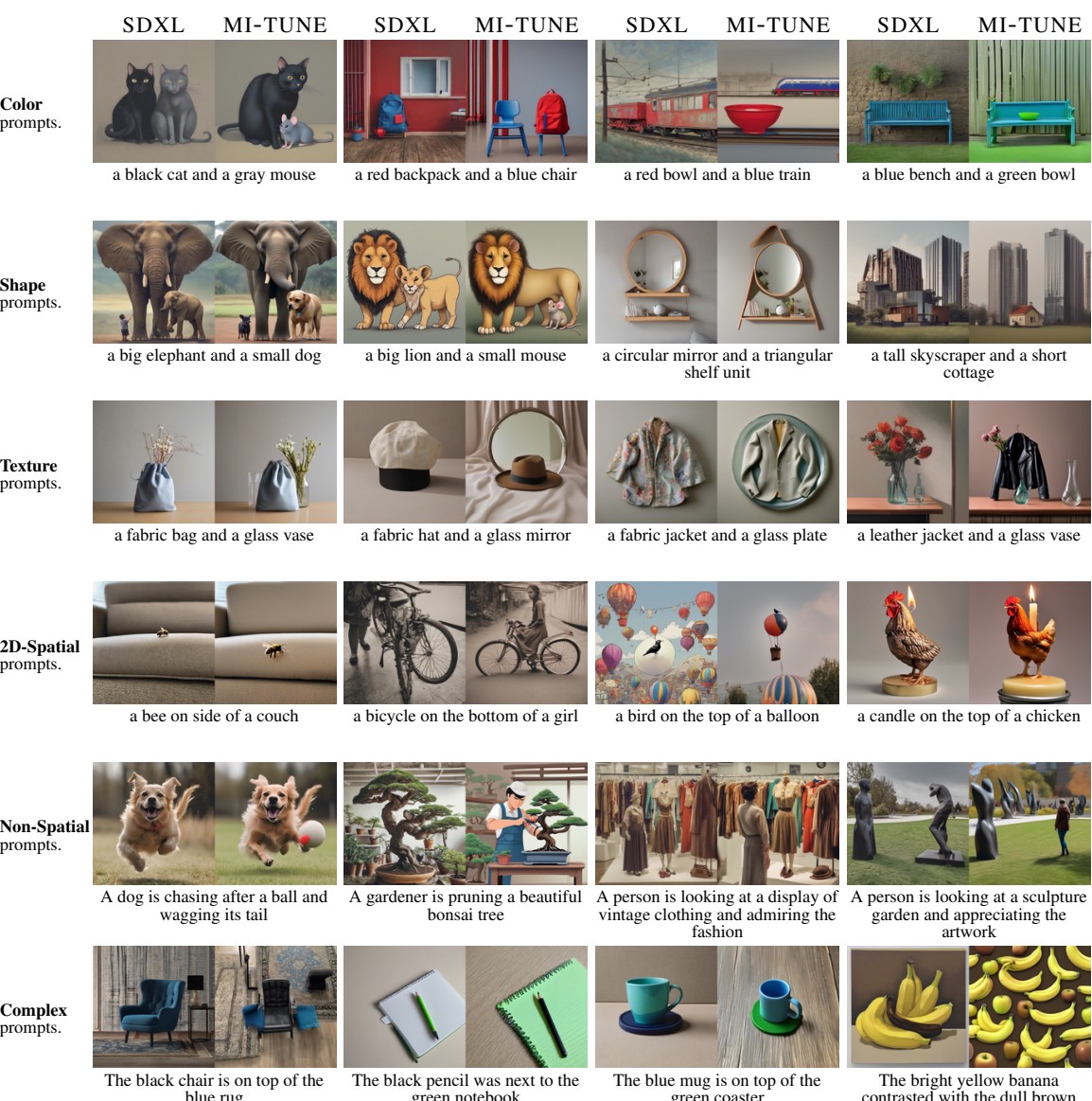

**Figure 14:** Qualitative examples from Table 4 (same seed used for a given prompt).

## H   FINE-TUNING WITH DIFFUSIONDB DATASET

### H.1   SELECTING IMAGES AND BLIP-VQA PROMPTS DECOMPOSITION

In this section, we provide additional details about using prompts created by real users, i.e., DiffusionDB.

**Dataset properties.** DiffusionDB was collected scraping the StableDiffusion discord channels "[...] *We download chat messages from the Stable Diffusion Discord channels with DiscordChatExporter, saving them as HTML files. We focus on channels where users can command a bot to run Stable Diffusion Version 1 to generate images by typing a prompt, hyperparameters, and the number of images* [...]" (Wang et al., 2022). The scraped data is then packaged into parquet files (containing metadata such prompt, image filenames and hyperparams) and zip files (containing the actual images in WebP format) and made available on HuggingFace.

**Fine-tuning with DiffusionDB.** We fine-tune SD-2.1-base on 1,250 prompts randomly sampled and compare two different scenarios. A first dataset is composed using images provided by DiffusionDB itself. As each prompt in DiffusionDB is paired to (about) 4 generated images we obtain a 5,000 prompt-image pairs reference dataset. For the second dataset, we use the 1,250 prompts to generate $M = 50$ images for each prompt and selecting the $k = 1$ image with the highest MI. We repeat this procedure 4 times to construct a complementary fine-tuning dataset with prompt-image 5,000 pairs. We fine-tune SD-2.1-base on each of the two datasets with pre-trained loss, then test on 500 DiffusionDB prompts (again, randomly selected and disjoint from the training set prompt-image pairs) generating 10 images for each test prompt.

**Table 14:** DiffusionDB.

| Model | HPS |
|---|---|
| SD-2.1-base | 23.99 |
| DiffusionDB | 24.35 |
| MI-TUNE | 25.32 |
| MI-TUNE ⊟ *base* | 1.33 |
| MI-TUNE ⊟ *DiffusionDB* | 0.97 |

*A* ⊟ B shows the abs. difference between A and B.

Table 14 (which is duplicating here Table 5 for simplicity) shows the results. Fine-tuning either using the DiffusionDB images or MI-TUNE can improve HPS score alignment with respect to the SD-2.1-base baseline. Yet, MI-TUNE improves upon using directly DiffusionDB images, i.e., using MI is very competitive compared to (expensive) manual labeling.

**BLIP-VQA prompts decomposition.** Our evaluation considers only HPS as we find that the higher prompt complexity does not well suit the BLIP-VQA prompt decomposition. Recall that BLIP-VQA requires to split the prompt into "noun phrases", each used to create a VQA for the BLIP model. Specifically, BLIP-VQA uses spaCy's English pipeline `en_core_web_sm` to extract noun phrases from the prompt which result complex when the prompt is complex. Below we report some examples related to extracting first three noun phrases extracted from human prompts.

Examples of good/easy segmentations:

- concept art of a silent hill monster. painted by edward hopper.
- anthropomorphic shark, digital art, concept art
- geodesic landscape, john chamberlain, christopher balaskas, tadao ando, 4k

Examples of segmentations with missing/broad subjects:

- a realistic architectural visualization of a sustainable mixed - use post - mordern post - growth walkable people oriented urban development.
- a realistic wide angle painting of a vintage cathode ray tube, in a park, in and advanced state of decay, psychedelic mushrooms all around, in a post apocalyptic city, ghibli, daytime, dynamic lighting
- render of dreamy beautiful landscape, dreamy, by herbaceous plants, artger, large scale, detailed vintage photo hyper realistic ultra realistic photo realistic photography, unreal engine, high detailed, 8 k

## H.2 Qualitative examples for DiffusionDB

| SD-2.1-base | Fine-tuned using DiffusionDB images | MI-TUNE | SD-2.1-base | Fine-tuned using DiffusionDB images | MI-TUNE |
|---|---|---|---|---|---|

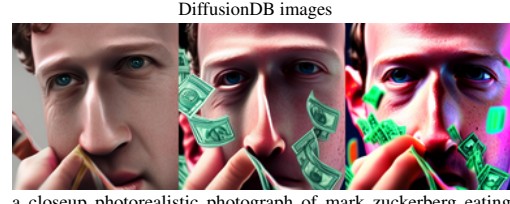

( ( ( ( a cute blue hedgehog with big gold ring and blue lightning in green grassland. ) ) ) ), big gold ring!, blue fur, clear sky, extremely detailed, fantasy painting, by jean - baptiste monge!!!!

a closeup photorealistic photograph of mark zuckerberg eating money. film still, vibrant colors. this 4 k hd image is trending on artstation, featured on behance, well - rendered, extra crisp, features intricate detail, epic compo

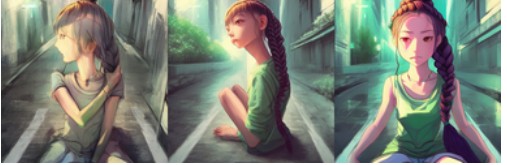

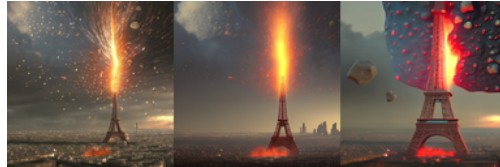

concept art, pretty girl sitting on street, braids blue and green, singular, junichi higashi, isamu imakake, intricate, balance, ultra detailed, full far frontal portrait, volumetric lighting, cinematic lighting + masterpiece

the eifel tower gets hit by an asteroid, multiple asteroids are in the air, paris in the background is burning, apocalyptic, highly detailed, 4 k, digital paintin, sharp focus, tending on artstation

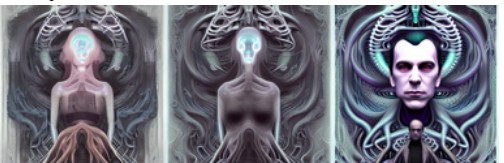

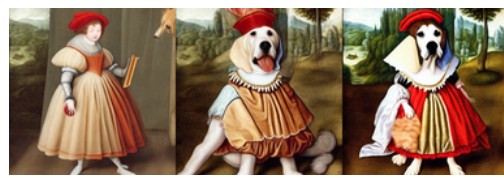

cosmic lovecraft giger fractal random antihero portrait, pixar style, by tristan eaton stanley artgerm and tom bagshaw.

a dog in a dress during the renaissance

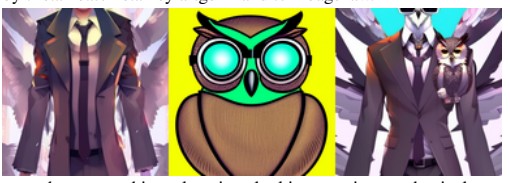

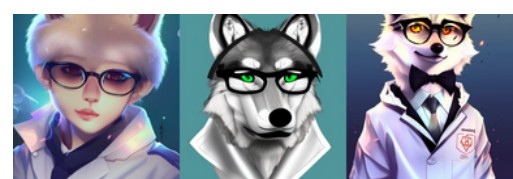

an anthropomorphic owl, serious looking wearing mechanical sunglasses and grey suit, by kawacy, trending on pixiv, anime, furry art, trending on furaffinity, mafia member.

anthropomorphic wolf with glasses wearing a lab coat, trending on artstation, trending on furaffinity, digital art, by kawacy, anime, furry art, warm light, backlighting, cartoon, concept art.

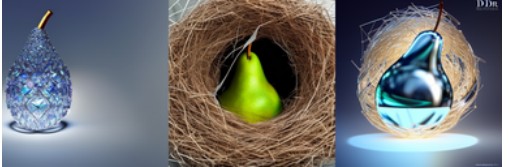

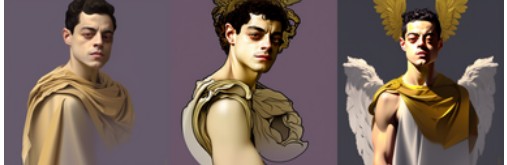

crystal big pear in a nest, transparent, with light glares, reflections, photo realistic, photography, photorealism, ultra realistic, intricate, detail, rim light, depth of field, unreal engine, dslr, rtx, style swarovski, dior, faberge.

rami malek as an angel in a golden toga, gray background, alphonse mucha, rhads, ross tran, artstation, artgerm, octane render, 1 6 k.

**Figure 15:** Qualitative examples from Table 5 (same seed used for a given prompt).

# I QUALITATIVE ANALYSIS OF MI AS AN ALIGNMENT MEASURE

Figure 16 is expanding Figure 1 to include qualitative examples for all categories in T2I-CompBench.

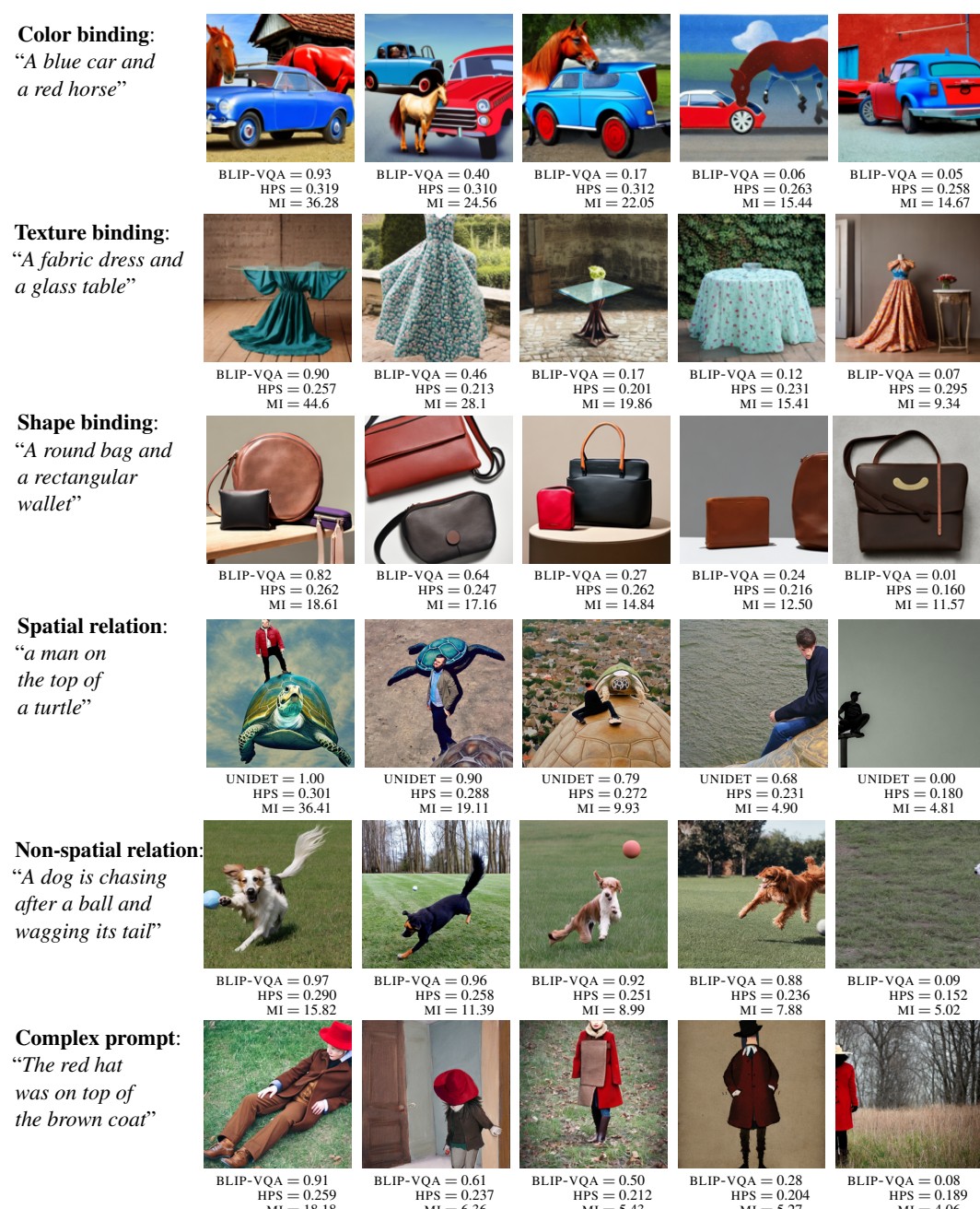

**Figure 16:** Qualitative analysis of MI as an alignment measure (all metrics decrease from left to right).

## J  BLIP-VQA, HPS AND MI SCORE DISTRIBUTIONS

**Figure 17:** CDF of alignment scores. Color reflect images rank based on MI.

The analysis presented in § 3.1 shows that BLIP-VQA, HPS and MI relate to each other. However, two aspects not discussed in § 3 are ($i$) the support of each metric and ($ii$) how the distribution of the scores compare between well and poor aligned images. In this ablation we address both aspects using the following protocol.

We considered all 700 training prompts for the color category (the consideration presented in this ablation extends to the other T2I-CompBench categories too), we generated 50 images for each prompt, and computed the 3 metrics for each of the 50 images. Last, for each prompt, we rank the images based on MI (1:highest, 50:lowest) – overall we obtained a 700 prompts × 50 images × 4 (3 metrics + 1 rank) tensor.

We then investigated if/how the MI rank affects the distribution of the scores for BLIP-VQA and HPS. Intuitively, given the highest-ranked (viz lowest-ranked) images based on MI, also BLIP-VQA and HPS should show very high values (viz low values). In practice, we first reordered the scores of the three metrics for each prompt based the MI rank and then we derived 50 distributions for each metric, one for each column in the tensor collecting the scores of each metric. Figure 17 shows the obtained distributions color coded based on the MI rank.

Considering the metrics support, we can notice a few differences among the three metrics. Specifically, BLIP-VQA is in the [0,1] range and for all rank values, the whole support is always used. Conversely, despite HPS is also in the [0, 1] range,[8] the actual support is more skewed – this corroborates the discussion presented in Appendix D. Last, while MI is unbounded, the scores are mostly contained in the [0-40] range.

Considering the relationship between the rank and the scores, all metrics show very similar patterns. Specifically, all distributions are very smooth no matter the rank. Moreover, as expected, for all metrics the distributions smoothly shift horizontally with respect to their rank – the color gradient separates very well red/high rank, yellow/middle rank, blue/low rank.

The kendal $\tau$ analysis reported in § 3.1 considers the 1st, 25th, 50th image for a prompt, selected by ranking the images based on their MI score. This is consistent with the analysis presented in Figure 17 and based on the figure we argue that our selection of 3 pictures (having the highest, mid, lowest scores for each prompt) is a reasonable choice for the results reported in § 3.1 as they are representative of the spectrum of values observed by the metrics.

---

[8]HPS is defined as the cosine similarity between image and text embeddings, similarly to CLIP. As such, theoretically, the score is in [-1, 1] range. However, in practice, and for the T2I-CompBench dataset, the score is effectively only in the [0, 1] range.

