# OpenReview forum: "Information Theoretic Text-to-Image Alignment"
_ICLR.cc/2025/Conference — ICLR 2025 Poster_

### Official Review · Reviewer_8y4V · 2024-10-27

**Soundness:** 3
**Presentation:** 3
**Contribution:** 3
**Rating:** 6
**Confidence:** 3

**Summary:**

This paper proposes MI-TUNE, a finetuning strategy for T2I models. This strategy introduces mutual information calculation into the T2I model tuning process to guide model alignment. In brief, the tuning method uses self-supervised fine-tuning and relies on a point-wise MI estimation between prompts and images to create a synthetic fine-tuning set for improving model alignment. This paper claims to be the first to introduce mutual-information calculation into T2I model training, leading to effective guidance for model alignment. The MI index is also measured and compared with well-established metrics and user study to prove the rationality. The experiments on MI are performed and evaluated on T2I-CompBench and user studies.

**Strengths:**

1.	This paper claims to be the first to introduce the mutual information (MI) for T2I model training alignment. And the MI is proven to be effective on model alignment and strongly agrees with well-established metrics.
2.	MI-TUNE results on several measurements (BLIP-VQA, HPS and Human) on image features reveals state-of-the-art performance.
3.	The paper also provides the proof for MI index to have a valid point-wise manner. Introducing mutual information into T2I model tuning is interesting, it may benefit the generative model training.
4.	The paper is easy to follow with algorithm and pseudo code provided.

**Weaknesses:**

1.	Note that the paper mainly focuses on SD-based (SD 2.1, SDXL) models. These models are mostly the same styles, e.g., similar network structures and traditional denoising training strategies. Is there any possibility that the MI tuning incorporated with flow-based models like DiT-based models (SD3, Pixart series or so). And it is interesting to see if the proposed MI tuning behaves different with different types of models.
2.	The evaluations on MI mainly focus on only simple semantic concepts like color, shape and texture. Is MI-tuning sensitive to object numbers or so?
3.	The paper fixes the denoising steps to 50 when inferencing an image, are there any differences in performance of MI-tuning when using different steps except 50?
4.	In quantitative analysis of Sect. 3.1, the paper mentions that the point-wise MI ranks images and select 1st, 25th and 50th as the representative images. Why the three images are representative? This needs more detailed explanations. Also, the reason of the selection needs quantitative analysis.
5.  Some of the ablations mentioned in previous sections are hard to locate in the following contents, the writing can be improved in this part.

**Questions:**

Please see the weakness part. How about the performance of MI Tuning on DiT based models or Flow-based model, since the flow-based model reveals stronger generation capability comparing to traditional DDPM training. If MI tuning strategy fails to have good performance on flow-based training, the impact may be weaker.

---

> ### Author Response · Authors · 2024-11-20
>
> We thank you for the comments. We reply to each point below.
>
> 1. *Note that the paper mainly focuses on SD-based (SD 2.1, SDXL) models. These models are mostly the same styles, e.g., similar network structures and traditional denoising training strategies. Is there any possibility that the MI tuning incorporated with flow-based models like DiT-based models (SD3, Pixart series or so). And it is interesting to see if the proposed MI tuning behaves different with different types of models.*
>
>     In the first comment, the reviewer is raising an interesting point regarding two complementary methodological aspects with respect to what presented in the submission, namely DiT models and flow-based models.
>
>     1. Considering DiT models, if they rely on DDPM (as Pixart), MI-TUNE can be seamlessly applied to them. To show this, we applied MI-TUNE to Pixart and we are ready to include additional results in the camera ready version of our paper. In the meanwhile, we will update our answer before the end of the discussion period, as soon as experiments terminate to share the new results.
>
>     2. Considering flow-based approaches, to fully exploit the self-contained nature of MI-TUNE (self-supervised fine-tuning, with no need for auxiliary models), it is important for the generative model to be amenable to estimating mutual information. While MI estimation with normalizing flows has recently appeared in the literature [1], the generative quality of such models is currently not on par with diffusion-based or rectified-flow based methods. MI estimation with rectified-flows (used by SD3 and Flux) is an open problem that, while very interesting, falls outside the scope of this work.
>
>         Indeed, adapting MI estimation would require re-deriving the time evolution of the mutual information along the generative dynamics of the deterministic rectified flows, and combine together the value of the learnt parametric drifts with score functions. In principle, such extension is within methodological reach, and its actual implementation will be an interesting avenue for future works.
>
>         To conclude, given a rectified-flow MI estimator, we believe that the MITUNE method we propose in this work could be seamlessly applied to such generative models, and contribute to improved text-to-image alignment.
>
>         [1] Butakov et al, “Mutual Information Estimation via Normalizing Flows”, NeurIPS 2024

---

> > ### Author Response · Authors · 2024-11-20
> >
> > 2. *The evaluations on MI mainly focus on only simple semantic concepts like color, shape and texture. Is MI-tuning sensitive to object numbers or so?*
> >
> >     The second comment regards the set of tasks/categories composing our evaluation. We acknowledge the comment and further elaborate on the following points:
> >
> >     1. First, it is probably reductive to refer to T2ICompBench as having only simple semantic categories. While Color, Shape, and Texture are indeed semantically related, the benchmark also includes Spatial (which is about the position of objects, e.g., “a balloon on top of a giraffe”), Non-Spatial (which is about actions, e.g., “a dog is walking on a leash with its owner”) and Complex (which is defined by the authors as a combination of multiple categories, e.g., “a black chair on the right of the wooden table”).
> >
> >     2. Second, our evaluation is going beyond T2I-CompBench with the evaluation on a set of prompts derived from DiffusionDB. This scenario was intentionally defined to complement the synthetic nature of T2I-CompBench with human-generated prompts, which express intricate semantics, beyond the categories considered in T2I-CompBench.
> >
> >     Ultimately, we agree with the reviewer that “numeracy” is another challenging T2I task. We are aware of a recent work [2] in the realm of inference-time methods, that tackles this challenge using attention map steering. Note that this approach requires an extremely fine-grained analysis of the prompt, as well as training an auxiliary model to predict the shape and location of a missing object, based on the layout of existing ones.
> >
> >     Next, for the “numeracy” task, current base models (SD2.1, SD3, and others) frequently fail in generating valid images, especially for high count numbers. The complexity of the “numeracy” task has been discussed in works such as [3, 4], and to the best of our knowledge, there are no widely accepted fine-tuning methods that tackle the numeracy task explicitly.
> >
> >     To offer a glimpse for the reviewer to appreciate an avenue for future work, we believe that MI-TUNE can be combined with inference-time methods. Indeed, the approach in [2] could be used to generate a fine-tuning set (that is, a set of images in which at least some have the correct number of objects), which can then be ranked according to MI, and used to fine-tune the base model of choice.
> >
> >     [2] Binyamin, et al., “Make It Count: Text-to-Image Generation with an Accurate Number of Objects” https://arxiv.org/pdf/2406.10210
> >
> >     [3] Cho, et al., “DALL-Eval: Probing the Reasoning Skills and Social Biases of Text-to-Image Generation Models”, in ICCV 2023 (https://arxiv.org/abs/2202.04053)
> >
> >     [4] Testonlin, et al., “Visual Enumeration is Challenging for Large-scale Generative AI”, https://arxiv.org/abs/2402.03328
> >
> >
> > 3. *The paper fixes the denoising steps to 50 when inferencing an image, are there any differences in performance of MI-tuning when using different steps except 50?*
> >
> >     The third comment regards the number of steps used by the diffusion models. This is indeed an interesting and relevant aspect. First of all, we note that using 50 steps is a common practice and is also the default setting for StableDiffusion. Now, changing this parameter (e.g., decreasing the number of steps) has the well-known effect of reducing image generation quality. In MI-TUNE, this also translates into a less accurate estimation of mutual information, which would be detrimental to our approach. Similarly, other approaches such as A&E would require considerable amendments: e.g., A&E uses 50 steps, splitting the denoising process into two phases, with only the first 25 steps acting on the attention maps. Any change in the number of steps has an impact on the “steering force” of attention maps, potentially harming the alignment process.

---

> > > ### Author Response · Authors · 2024-11-20
> > >
> > > 4. *In quantitative analysis of Sect. 3.1, the paper mentions that the point-wise MI ranks images and select 1st, 25th and 50th as the representative images. Why the three images are representative? This needs more detailed explanations. Also, the reason of the selection needs quantitative analysis.*
> > >
> > >     The fourth point regards the selection we operated when presenting the results in Sec 3.1. We thank the reviewer for pointing out this aspect and to better address the comment we added a new Appendix J to our revised paper where we analyze in detail the score distributions for BLIP-VQA, HPS and MI. All distributions exhibit a similar trend, and we notice that they are not significantly skewed. As a consequence, selecting the 1st, 25th and 50th images as representative samples is a reasonable choice. Additional details and plots for the cumulative distribution function of the score is available in the new appendix.
> > >
> > > 5. *Some of the ablations mentioned in previous sections are hard to locate in the following contents, the writing can be improved in this part.*
> > >
> > >     The last comment suggests to improve the editorial exposition of the appendices. We acknowledge the comment, as indeed our manuscript is quite dense. To better scope the different ablations, we revised the manuscript so that the appendices table of content provides a summary description (in light blue) of each ablation and reference to the main paper.

---

> ### Comment · Reviewer_8y4V · 2024-11-25
>
> Thanks for the detailed responses from the authors. Part of my concerns are solved. However, some of the points regarding the experiments on rectified-flow-based models remains unsolved, which makes me feel that the strategy strongly depends on the DDPM-style learning (unknown performance to flow-based model) and may weaken the contribution the paper should have made. Overall, it is still good to know that using mutual information to improve the DDPM-based model learning on text-to-image alignment for image generation and I will keep my score after considering the detailed response and other comments.

---

> > ### Author Response · Authors · 2024-11-25
> >
> > Thank you for your comment, we’re glad some of your concerns were addressed.
> > The methodology we use to estimate mutual information has been applied also to score-based diffusion models [Franzese et.al, ICLR 2024, Kong et al., ICLR 2024], so it is not restricted to DDPM-only generative models. As discussed in our previous answer, we think adapting the method to estimate MI using rectified-flows is within reach. Once available, it can be readily used by the MITUNE strategy. We think however that this could be entitled as a new methodological contribution alone, and should be addressed separately from the MITUNE approach. Thank you very much for the new perspectives this discussion has opened!

---

> > ### Author Response · Authors · 2024-11-25
> > **MI-TUNE on PixArt-sigma**
> >
> > Please find below a summary of an evaluation of MI-TUNE on T2I-Compbench on Pixart-$\Sigma$ [1]. For these new results we used the same protocol described in our paper. Namely, we used the 700 prompts available for a category, to fine-tune Pixart-$\Sigma$ using MI-TUNE (the fine-tuning set is composed by generating 50 images for a prompt, and selecting the image with the highest MI score). Then, we evaluated the fine-tuned and the base Pixart-$\Sigma$ models using the predefined 300 test prompts for the category (generating 10 images for each prompt). Given the limited time available, we managed to collect results only for the color and shape categories, and report the BLIP-VQA scores in the table below.
> >
> > | model                 | color | shape |
> > |---------------------- | ---- | ----- |
> > |Pixart-$\Sigma$  | 59.61 | 47.98 |
> > |MI-TUNE            | 64.33 | 52.35 |
> >
> > These new results corroborate our claim that MI-TUNE does not depend on specific model architectures, and that it can be used to improve alignment of a variety of base models (all in all, we have now results on SD2.1-base, SDXL, Pixart-$\Sigma$).
> >
> > We will complete the benchmark with the remaining categories for the camera ready.
> >
> > [1] https://huggingface.co/PixArt-alpha/PixArt-Sigma-XL-2-1024-MS

---

### Official Review · Reviewer_LaKZ · 2024-11-02

**Soundness:** 2
**Presentation:** 2
**Contribution:** 3
**Rating:** 6
**Confidence:** 4

**Summary:**

The proposed MI-TUNE introduces a self-supervised fine-tuning approach to enhance text-to-image alignment in diffusion models. At its core, the method leverages Mutual Information (MI) between text prompts and their corresponding generated images to improve model alignment, eliminating the need for human annotation. The method generates multiple images per prompt, selects the top-K aligned ones based on MI estimation, and uses these for fine-tuning.

**Strengths:**

1. The main problem "Is mutual information meaningful for alignment?" is compelling and necessary, showing the potential of MI as a new direction for text-image alignment
2. The paper is well-organized and easy to follow.
3. The proposed fine-tuning approach is intersting and seems to have predictive power.

**Weaknesses:**

1. MI scores are missing from comparison tables and images despite being central to the method.
2. Figure 1 only demonstrates MI effectiveness on simple category prompts (color, texture, shape), lacking validation on more challenging cases like spatial relationships or complex compositions

**Questions:**

1. How were the 700 prompts selected for the MI quantitative analysis?

---

> ### Author Response · Authors · 2024-11-20
>
> We thank you for the comments. We reply to each point below
>
> 1. *MI scores are missing from comparison tables and images despite being central to the method.*
>
>     The first comment regards the lack of using MI as an evaluation metric in the benchmark. **We do not report MI scores as an evaluation metric in the comparison tables, because using MI for both fine-tuning set selection and evaluation would represent an unfair advantage for MI-TUNE.**
>
>     This would be the same scenario GORS authors (Huang et al., 2023) call “biased evaluation”, i.e., the metric used for evaluation is also used during fine-tuning. To be even more precise, GORS results in Table 1 represent a “biased evaluation”, as this mechanism is an intrinsic aspect of the GORS method itself. That said, even in this condition, GORS presents inconsistent performance and often does not even result to be the best alternative fine-tuning method.
>
>     **For additional information about MI scores computed on generated images in our qualitative analysis (see Fig. 1), we added a new appendix J** with a discussion of distribution of such scores.
>
> 2. *Figure 1 only demonstrates MI effectiveness on simple category prompts (color, texture, shape), lacking validation on more challenging cases like spatial relationships or complex compositions*
>
>     The second comment regards missing categories from Fig.1. The reviewer is right in pointing out that Fig.1 could have been exhaustive and include all categories. However, this was a mere editorial choice, dictated by lack of space, and to underline that Fig.1 was reporting only a **qualitative analysis**.
>
>     In other words, Fig.1 motivates our intuition that MI is a meaningful signal for alignment. Sec 3.1. further corroborates this, with more data analysis and points to a user study detailed in Appendix B.1.
>
>     That said, **ultimately, the final assessment of MI is carried over in Table 1 with the actual comparison against all alternative methods (including HPS and Human evaluation too), with additional results about SDXL (Table4) and DiffusionDB (Table5).**
>
>     For completeness, in **Appendix I of the revised manuscript we added example figures for all categories** complementing the one reported in Fig.1. The new figures show the same trend as the one already provided in the original Fig.1.
>
> 3. *How were the 700 prompts selected for the MI quantitative analysis?*
>
>     The third comment relates to the train/test composition of the datasets we used. Based on the reviewer’s question, we realize that some clarifications are in order, thank you!
>
>     Our evaluation considers two datasets, namely T2I-CompBench and DiffusionDB.
>
>     T2ICompBench pre-defines 700/300 train/test prompts for each category. So, to answer the reviewer’s question, **the 700 prompts we use are those that are predefined by T2I-CompBench.** In our evaluation, we use the pre-defined splits (to foster replicability and comparison against other studies using this same dataset). Quoting T2I-CompBench authors
>
>     [...]*we introduce three categories and six sub-categories of compositionality, attribute binding (including three sub- categories: color, shape, and texture), object relationships (including two sub-categories: spatial relationship and non-spatial relationship), and complex compositions. We generate 1,000 text prompts (700 for training and 300 for testing) for each sub-category, resulting in 6,000 compositional textprompts in total. We take the balance between seen v.s. unseen compositions in the test set, prompts with fixed sentence template v.s. natural prompts, and simple v.s. complex prompts into consideration when constructing the benchmark. The text prompts are generated with either predefined rules or ChatGPT [45], so it is easy to scale up*[...]
>
>     Conversely, for DiffusionDB, we randomly sampled 5,000/1,250 (train/test) prompt-image from the dataset. We’ll make available our selection as part of the code artifacts, in an effort to foster reproducibility of our results.
>
>     In the revised manuscript we clarified this in section 3.1.

---

> > ### Comment · Reviewer_LaKZ · 2024-11-22
> >
> > Thank you for your thoughtful comments, which help detail understanding.
> >
> > However, I'm still unclear about why the MI scores are not reported. As you mentioned, the author of GORS referred to this evaluation as "biased evaluations," yet they presented both biased and unbiased evaluation results in their work. Thus, including both types of results would not be inappropriate for comparison. I just wonder whether the MI scores improve after applying the MI-Tune method. Based on the method's premise, I would expect the MI scores to be enhanced following training. Reporting these results would help clarify the effectiveness of the approach.

---

> > > ### Author Response · Authors · 2024-11-25
> > > **MI scores**
> > >
> > > Thank you for your patience! We are sharing the results on MI scores that you asked below.
> > > In both tables, one on color and the other on shape categories, we report the average and main quantiles for the Mutual Information score, comparing our baseline (SD2.1-base) against our proposed method MITUNE. Specifically, we used the “test configuration” in the T2I-CompBench benchmark: we used 300 test prompts, and generated 10 images per prompt.
> > >
> > > Our results clearly indicate a substantial improvement for the Mutual Information as a result of our MI-TUNE strategy, when compared to the baseline. This is in-line with our expectations and with the reviewer’s intuition. Thank you for helping us further strengthen our work!
> > >
> > > Color Attribute :
> > > | Method | MI (average) | MI (0.25 quantile) | MI (0.5 quantile) | MI (0.75 quantile) |
> > > |---|---|---|---|---|
> > > | SD2.1-base | 20.48 | 12.56 | 17.46 | 24.68 |
> > > | MITUNE | 42.09 | 24.97 | 35.63 | 55.50 |
> > >
> > >
> > > Shape Attribute :
> > > | Method | MI (average) | MI (0.25 quantile) | MI (0.5 quantile) | MI (0.75 quantile) |
> > > |---|---|---|---|---|
> > > | SD2.1-base | 18.68 | 12.21 | 15.61 | 21.04 |
> > > | MITUNE | 26.43 | 18.58 | 23.49 | 30.85 |

---

> > > > ### Comment · Reviewer_LaKZ · 2024-11-26
> > > >
> > > > Thanks to the author for the detailed reply, which solved my concerns. Then, I decide to raise my rating.

---

### Official Review · Reviewer_8jmm · 2024-11-05

**Soundness:** 2
**Presentation:** 2
**Contribution:** 2
**Rating:** 6
**Confidence:** 4

**Summary:**

This paper introduces Mutual Information (MI) to guide model alignment, which uses self-supervised fine-tuning manner. It relies on a point-wise MI estimation between prompts and images to create a synthetic fine-tuning set for improving model alignment.

**Strengths:**

- The idea of introducing  self-supervised fine-tuning manner is interesting.

- Mutual Information in the pipeline is simple and effective.

- It seems is a plug-and-play module, which is useful for most T2I models.

**Weaknesses:**

- More detailed ablations are needed. The authors employ MI as the metric to select fine-tuning samples, which eliminates the extra usage of other models. However, what if we use SOTA VQA models as the metric? Intuitively, SOTA VQA models are more precise than the MI metric.

- An inherent drawback of MI is that it can measure how much help comes from the prompt but cannot guide in the right direction. For example, in cases of color misalignment, how should we deal with this issue?

**Questions:**

Please see the waeakness.

---

> ### Author Response · Authors · 2024-11-20
>
> We thank you for the comments. We reply to each point below.
>
> * *More detailed ablations are needed. The authors employ MI as the metric to select fine-tuning samples, which eliminates the extra usage of other models. However, what if we use SOTA VQA models as the metric? Intuitively, SOTA VQA models are more precise than the MI metric.*
>
> 1. The first comment regards the evaluation protocol used. We agree with the reviewer that the fine-tuning set composition and the evaluation metric are important. In this regard, we highlight the following points:
>
>     1. First of all, using BLIP-VQA for fine-tuning set composition AND final evaluation leads to biased results. This is demonstrated by GORS authors (Huang et al., 2023) as we mentioned in Sec.4.4 of the original submission.
>
>
>         Note also that **the results for GORS reported in Table1 of our paper already reflect the experiment suggested by the reviewer (BLIP-VQA is used both during fine-tuning of GORS and its evaluation)**, as this an integral mechanism of GORS. Yet, GORS performance is inconsistent across categories and it often does not even result as the best method within the fine-tuning family.
>
>
>     2. **We also consider alternative approaches to the use of MI for composing the fine-tuning set. Table 3** presents an ablation considering HPS (as a possible alternative alignment metric) as well as a mix of generated and real pictures.
>
>     3. Finally, **Table 1 also includes a Human study** where real people were asked to provide alignment feedback (details of the survey campaigns are available in Appendix B). While the surveys were done on a relatively small set of users, they are still relevant to investigate the meaningfulness of MI to steer model alignment.
>
>
> * *An inherent drawback of MI is that it can measure how much help comes from the prompt but cannot guide in the right direction. For example, in cases of color misalignment, how should we deal with this issue?*
>
> 2. The second comment is focusing on the role/importance of MI. We agree with the reviewer in saying that MI can measure “how much help” comes from the prompt. Indeed, it helps the model to better denoise the generated images, during fine-tuning.
>
>     However, “guiding the model in the right direction” is what inference-time methods such as A&E do, while MI-TUNE is a fine-tuning method. Abstracting from our evaluation, we are comparing the two alternatives: i) an implicit guidance through fine-tuning vs. ii) an explicit guidance at inference time by acting (for instance) on cross-attention maps.
>
>     Taking color misalignment as a case study, as suggested by the reviewer, we show below that our method yields the best results (taken from Table 1 in the paper).
>
>     | Method | BLIP-VQA |
>     |---|---|
>     | SD2.1-base | 49.65 |
>     | A&E | 61.43 |
>     | MI-TUNE | 65.04 (+5.88% wrt A&E) |

---

> > ### Comment · Reviewer_8jmm · 2024-11-25
> >
> > Thanks for the response. After considering the response and other reviews, I have decided to maintain my current rating.

---

### Official Review · Reviewer_A9zb · 2024-11-05

**Soundness:** 3
**Presentation:** 3
**Contribution:** 3
**Rating:** 6
**Confidence:** 4

**Summary:**

This work proposes a self-supervised approach using Mutual Information (MI) for model alignment, requiring only the pre-trained T2I model and a simple fine-tuning process, outperforming current state-of-the-art methods.

**Strengths:**

1. The proposed method does not require additional image datasets for training.
2. The idea is relatively novel.

**Weaknesses:**

1. Comparison methods, "Attend and Excite (A&E) (Chefer et al., 2023b), Structured Diffusion Guidance(SDG) (Feng et al., 2023b) and Semantic-aware Classifier-Free Guidance (SCG) (Shen et al., 2024)"  mentioned in line 329, is implemented on SD1.4, but the proposed work is implemented on SD 2.1, which is powerful than SD1.4 and may introducing evaluation bias.
2. The experiments, the train set, and the test set are split from the same dataset, but this may exist some correlations, what's the results on out-of-distribution prompts?
3. Although mutual information provides a theoretical explanation, it essentially uses classifier guidance as a loss for fine-tuning. Given that this approach is so simple, has there been any similar attempt in previous work?

**Questions:**

Referring Weakness

---

> ### Author Response · Authors · 2024-11-20
>
> We thank you for the comments. We reply to each point below.
>
>
> *1. Comparison methods, "Attend and Excite (A&E) (Chefer et al., 2023b), Structured Diffusion Guidance(SDG) (Feng et al., 2023b) and Semantic-aware Classifier-Free Guidance (SCG) (Shen et al., 2024)" mentioned in line 329, is implemented on SD1.4, but the proposed work is implemented on SD 2.1, which is powerful than SD1.4 and may introducing evaluation bias.*
>
> The first comment raises attention to model version and fairness of the comparison. We acknowledge the need for a fair comparison and believe the comment is due to insufficient clarity in the description of the setup used in our evaluation.
>
> To clarify, for results of the 6 alternative methods reported in Table 1, **we did NOT report results from the related papers. Rather, we ran those methods ex-novo on the benchmark, testing them all on SD 2.1** – this was the only way to guarantee a fair comparison.
>
> We clarified this in Sec. 4.3 by adding
>
> *We underline that since results reported in the literature for both families do not necessarily refer to the same base models, to guarantee a fair comparison, we adapted and evaluated all methods on SD-2.1-base.*

---

> > ### Author Response · Authors · 2024-11-20
> >
> > *2. The experiments, the train set, and the test set are split from the same dataset, but this may exist some correlations, what's the results on out-of-distribution prompts?*
> >
> >
> > The second comment raises attention to out-of-distribution testing. This is an interesting and valuable point. There are two aspects to consider:
> >
> > 1. **T2ICompBench test set already contains a mix of in- and out-of-distribution adjective-noun pairs.** Quoting T2ICompBench authors (Huang et al., 2023)
> >
> >     [...]*the 300-prompt test set of each sub-category consists of 200 prompts with seen adj-noun compositions (adj-noun compositions appeared in the training set) and 100 prompts with unseen adj-noun compositions (adj-noun compositions not in the training set)*[...]
> >
> > 2. Some of the results in Appendix E.3 also relate to an out-of-distribution scenario. Specifically, since MI-TUNE requires to fine-tune a model for each category, in **Appendix E.3** of the original submission we reported a study investigating if it was possible to combine different per-category models into a single “surrogate” model that could be used for all categories at once – basically this would ease deployment and usability.
> >
> >     The link between this analysis and the reviewer comment is in **Fig.7** which shows the performance of a scenario where the color and shape categories are combined into a single model with a weighting $\lambda$: **when $\lambda=0$ the performance in the figure represents the shape-only model used for generating images for the color category prompts (the reverse for $\lambda=1$)** – this is a more rigid out-of-distribution setting with respect to the one natively provided in T2ICompBench.
> >
> >     To better appreciate this scenario, we report here below the relevant performance scores.
> >
> >     |                   | test on color | test on shape |
> >     |---                |---            |---            |
> >     | finetune on color | 61.57         | 47.83         |
> >     | finetune on shape | 59.5          | 48.40         |
> >     | SD-2.1-base       | 49.65         | 42.71         |
> >
> >
> >
> >
> >     Note that, for simplicity and due to time constraints, the results in this ablation uses R=1, while Table 1 reports the best observed results (R=3).
> >
> >     As expected, the out-of-distribution tests highlight a performance drop compared to in-distribution settings. Yet, even in this scenario performance is much improved with respect to the baseline.

---

> > > ### Author Response · Authors · 2024-11-20
> > >
> > > *3. Although mutual information provides a theoretical explanation, it essentially uses classifier guidance as a loss for fine-tuning. Given that this approach is so simple, has there been any similar attempt in previous work?*
> > >
> > > The third point regards the comparison of the proposed method with respect to available literature. First of all, we shared the same surprise with the reviewer when realizing that the simple approach we propose –  where we consider simplicity as a virtue of a practical method – had not been attempted before.
> > >
> > > To be more precise, there are 2 key ingredients in our approach: self-supervision (to generate a set of images) and fine-tuning set composition (based on an automatic selection of the generated images). Among the two, **the use of a theoretically sound measure like MI between prompt and images is (in our opinion) the not obvious step, yet advantageous based on our evaluation.**
> > >
> > > As mentioned at the end of Sec 3.1, some very recent literature indirectly suggests the possible use of MI for alignment. Despite this, to the best of our knowledge, no previous work includes the two ingredients mentioned above, nor explicitly investigates MI for alignment. This is a quote from Sec.3.1
> > >
> > > [...] *Kong et al. (2024) estimates pixel-wise mutual information between natural prompts and the images generated at each time-step of a backward diffusion process. They compare such “information maps” to cross-attention maps (Tang et al., 2023) in an experiment involving prompt manipulation – modifications of the initial prompt during reverse diffusion – and conclude that MI is much more sensitive to information flow from prompt to images. In a similar vein, Franzese et al. (2024) compute MI between prompt and images at different stages of the reverse process of image generation. Experimental evidence indicates that MI can be used to analyze various reverse diffusion phases: noise, semantic, and denoising stages (Balaji et al., 2022a). While previous studies do not explicitly focus on alignment, they indirectly support our intuition that MI estimated using a diffusion model gauges the amount of information a text prompt conveys about an image (and vice-versa) which is key for T2I alignment.* [...]

---

> > > > ### Comment · Reviewer_A9zb · 2024-11-26
> > > >
> > > > Thank you for your response. After careful consideration, I believe the previous score already reflects the quality of the article accurately, so I will keep the score unchanged.

---

### Author Response · Authors · 2024-11-20

We thank the reviewers for their work in assessing our paper, and for the helpful comments and questions. Thank you for noting the originality of our approach, that blends self-supervision and pointwise mutual information estimation, and for recognizing that our method is “plug-and-play”, as it does not need auxiliary procedures for text-to-image alignment. Reviewers also duly noted the new state-of-the-art results we obtained on both standard benchmarks and on difficult real-world data.

We have prepared a detailed answer to all comments and questions, hoping that our clarifications will lead to an even more positive assessment of our work. We also uploaded a revised version to include the following:

* Clarified in Sec.3 that we use all prompts from T2I-Compbench
* Added Appendix I to extend Fig.1 by including examples from all T2I-CompBench categories, and updated Fig.1 caption to refer to the new appendix.
* Clarified in Sec.4.3  that results in Table 1 refer to evaluating all methods on SD2.1
* Added Appendix J to discuss the relationship between metrics distributions and rank
* Reworked the appendix table of content to ease the understanding of the scope of each appendix and reference them to the relevant parts of the main paper.
* We will integrate additional results of MI-TUNE applied to a DiT-based architecture, specifically PixArt, as suggested by one reviewer. Experiments are in progress, and we will provide an update message as soon as they finish.

---

> ### Author Response · Authors · 2024-11-25
>
> Please find the following additional results:
>
> 1. The Mutual Information scores of baseline (SD2.1-base) against our proposed method MI-TUNE, in the answer to Reviewer LaKZ.
>
>     These results clearly indicate a substantial improvement for the Mutual Information as a result of our MI-TUNE strategy, when compared to the baseline.
>
> 2. The additional result of MI-TUNE on the DiT-based PixArt-$\Sigma$, in the answer to Reviewer 8y4V.
>
>     These results corroborate our claim that MI-TUNE does not depend on specific model architectures, and that it can be used to improve alignment of a variety of base models (all in all, we have now results on SD2.1-base, SDXL, PixArt-$\Sigma$).
>
> Thanks to the reviewers for helping us further strengthen our work!

---

### Meta-Review · Area_Chair_1ZMe · 2024-12-18

**Metareview:**

"This paper proposes a method base on mutual information to improve text-to-image alignment in generative text-to-image generative models.
The strengths of the paper indicated in the reviews include: well written, simple finetuning process that does not require additional data or models to improve performance, novel idea & interesting direction, and the proposed method can be combined with many T2I models.
Weaknesses: lacks evaluation on out-of-distribution prompts, misses some comparisons, MI scores not reported while central part of method, validation limited to relatively simple prompts,  study limited to Unet diffusion models and missing evaluation of transformer-based models with flow training, missing ablations on nr of inference steps.

**Additional Comments On Reviewer Discussion:**

In response to the reviews the authors have submitted a detailed rebuttal and a revised version of the manuscript. The author response has addressed most reviewer comments, and the reviewers unanimously recommend accepting the paper. The AC follows the recommendation as no major outstanding concerns remain after the author-reviewer discussion."

---

### Decision · Program_Chairs · 2025-01-22

Accept (Poster)